# Niche Tet maintains germline stem cells independently of dioxygenase activity

Renjun Tu [1,5], Zhaohua Ping [2,5], Jian Liu[3], Man Lung Tsoi[1,4], Xiaoqing Song[2], Wei Liu[3] & Ting Xie [1,2]✉

## Abstract

**Ten-eleven translocation (TET) proteins are dioxygenases that convert 5-methylcytosine (5mC) into 5-hydroxylmethylcytosine (5hmC) in DNA and RNA. However, their involvement in adult stem cell regulation remains unclear. Here, we identify a novel enzymatic activity-independent function of Tet in the *Drosophila* germline stem cell (GSC) niche. Tet activates the expression of Dpp, the fly homologue of BMP, in the ovary stem cell niche, thereby controlling GSC self-renewal. Depletion of Tet disrupts Dpp production, leading to premature GSC loss. Strikingly, both wild-type and enzyme-dead mutant Tet proteins rescue defective BMP signaling and GSC loss when expressed in the niche. Mechanistically, Tet interacts directly with Bap55 and Stat92E, facilitating recruitment of the Polybromo Brahma associated protein (PBAP) complex to the dpp enhancer and activating Dpp expression. Furthermore, human TET3 can effectively substitute for *Drosophila* Tet in the niche to support BMP signaling and GSC self-renewal. Our findings highlight a conserved novel catalytic activity-independent role of Tet as a scaffold protein in supporting niche signaling for adult stem cell self-renewal.**

**Keywords** Germline Stem Cells; Jak-Stat Signaling; Self-renewal and Differentiation; Stem Cell Niche; Tet
**Subject Categories** Chromatin, Transcription & Genomics; Signal Transduction; Stem Cells & Regenerative Medicine

## Introduction

Ten-eleven translocation (TET) family proteins, including TET1, TET2, and TET3 in mammals, are the dioxygenases that can convert 5-methylcytosine (5mC) in DNA and RNA to 5-hydroxymethylcytosine (5hmC) (Delatte et al, 2016; Wu and Zhang, 2017). For DNA, Tet proteins can progressively oxidize 5mC to 5hmC, 5-formylcytosine (5fC), and 5-carboxylcytosine (5caC), which eventually leads to DNA demethylation (Wu and Zhang, 2017; Yang et al, 2020). They control the epigenetic regulation of gene expression, which is important for various biological processes, including embryonic stem cell differentiation (Wu and Zhang, 2017; Yang et al, 2020). Only recently, Tet has been shown to catalyze 5mC to 5hmC in RNA in *Drosophila* and mouse (Delatte et al, 2016; Lan et al, 2020). In addition to the oxidization of 5mC in DNA and RNA, they can also have dioxygenase-independent functions in the nucleus, including stabilizing transcription factors and regulating histone modifications (Wang et al, 2017; Yang et al, 2020).

In *Drosophila*, only one Tet exists and is closely related to human TET1 and TET3 (Fig. EV1A). Tet has first been reported to demethylate N[6]-methyladenosine (m6A) in DNA in *Drosophila* and is expressed in the germline for controlling early germline stem cell (GSC) progeny differentiation (Zhang et al, 2015). However, another study argued that Tet is undetectable in germ cells of the adult ovary and dispensable for GSC progeny differentiation (Wang et al, 2018). In addition, it has been further demonstrated to interact with a Polycomb protein to activate gene transcription in neurons of the *Drosophila* brain by demethylating 6 mA in DNA (Yao et al, 2018). *Drosophila* Tet has also been shown to control the conversion of 5mC in RNA into 5hmC (Delatte et al, 2016), and it is required in midline glia to regulate axon guidance in the developing brain and control glial homeostasis in the adult brain by controlling 5hmC in mRNA (Frey et al, 2022; Ismail et al, 2019; Singh et al, 2023).

Each *Drosophila* ovary contains approximately 12–16 ovarioles, in which the germarium harboring two or three GSCs is located the most anterior. GSCs continuously divide to generate cystoblasts (CBs), which then form mitotic cysts (2-cell, 4-cell, and 8-cell) and 16-cell cysts through synchronous incomplete cytokinesis (Fig. EV1B) (Spradling et al, 1997; Villa-Fombuena et al, 2021; Xie, 2013). In the germarium, two or three GSCs directly contact cap cells and anterior inner germarial sheath cells (IGS cells, also known as escort cells), forming the niche that regulates GSC self-renewal. On the other hand, CBs, mitotic cysts, and early 16-cell cysts are enveloped by the elongated cellular processes of middle and posterior IGS cells. These IGS cells create the niche that promotes the differentiation of GSC progeny (Kirilly et al, 2011; Wang et al, 2011; Xie and Spradling, 2000). Moreover, these IGS cells form multiple niche compartments interacting with mitotic cysts and 16-cell cysts separately to control their stepwise development (Tu et al, 2021). Cap cells produce BMP-like Dpp

[1]Division of Life Science, The Hong Kong University of Science and Technology, Clear Water Bay, Kowloon, Hong Kong Special Administrative Region, China. [2]Stowers Institute for Medical Research, 1000 East 50th Street, Kansas City, MO, USA. [3]Shenzhen Key Laboratory for Neuronal Structural Biology, Biomedical Research Institute, Shenzhen Peking University-The Hong Kong University of Science and Technology Medical Centre, Shenzhen, Guangdong, China. [4]Centre for Regenerative Medicine and Health, Hong Kong Institute of Science and Innovation, Chinese Academy of Sciences, New Territories, Hong Kong Special Administrative Region, China. [5]These authors contributed equally: Renjun Tu, Zhaohua Ping.
✉E-mail: tgx@ust.hk

for directly signaling to GSCs for controlling their self-renewal by repressing the expression of key differentiation factor Bam and thus preventing differentiation (Chen and McKearin, 2003; Diaz-Torres et al, 2021; Rojas-Rios et al, 2012; Song et al, 2004; Xie and Spradling, 1998). In addition, cap cells also use E-cadherin-mediated cell adhesion to anchor GSCs for long-term self-renewal (Song et al, 2002). Furthermore, Jak-Stat signaling controls Dpp expression in cap cells to maintain GSC self-renewal (Lopez-Onieva et al, 2008; Wang et al, 2008). However, it remains largely unclear how Jak-Stat signaling transcriptionally controls *dpp* expression at the molecular level.

This study has revisited the role of Tet in the regulation of GSC development in the *Drosophila* ovary. We have used RNA in situ hybridization and a EGFP knock-in strain to show that Tet is only expressed in somatic niche cells but not germ cells, and its depletion from the niche leads to GSC loss due to the down-regulation of Dpp. We have further demonstrated that the dioxygenase activity of Tet is dispensable in the niche for controlling GSC self-renewal and instead Tet functions as a protein bridge to connect the PBAP complex and Stat92E in the niche to activate *dpp* expression and maintain GSC self-renewal. Finally, human TET3 is sufficient to replace Tet function in the niche to maintain BMP signaling and GSC self-renewal. Therefore, we have uncovered a novel functional mode of Tet independent of its enzymatic activity in the niche to control stem cell self-renewal, which enriches our understanding of TET functions in tissue development, stem cell regulation, and tumorigenesis. Our study has contributed to ruling out the possibility that Tet is expressed in GSC progeny for controlling their differentiation in the *Drosophila* ovary.

## Results

### Tet is required in the niche for controlling GSC self-renewal

To investigate the function of Tet in the regulation of GSCs in the *Drosophila* ovary, we first examined the expression patterns of *Tet* at both the mRNA and protein level in the germarium. We used RNA hybridization chain reaction-fluorescence in situ hybridization (HCR-FISH) to show that *Tet* mRNA restricts its expression to somatic cells in the adult ovary, including terminal filament cells, cap cells, IGS cells, and follicle cells (Fig. 1A). To further detect Tet protein expression in the germarium, the Flag-EGFP dual tag is knocked into the C-terminal end of the coding region of the *Tet* gene to express a C-terminally Flag-EGFP tagged Tet protein. Consistent with *Tet* mRNA expression pattern, Tet-EGFP protein is also expressed in the nucleus of somatic niche cells, including cap cells and IGS cells, but not in germ cells (Figs. 1B and EV1C). These results suggest that Tet restricts its mRNA and protein expression to somatic cells in the adult *Drosophila* ovary, including GSC niche cells. Our findings support the previous conclusion that Tet is not expressed in GSC progeny for controlling their differentiation at adult stage (Wang et al, 2018).

Since Tet is expressed in niche cells, we then proceeded to investigate its function in the niche by performing knockdown using *bab1-Gal4*, a widely used Gal4 driver enriched in the niche (Bolivar et al, 2006; Diaz-Torres et al, 2021). In adult stages,

*bab1-Gal4* exhibits high expression in terminal filament cells and cap cells, with low-level expression observed in some anterior IGS cells (Bolivar et al, 2006). To achieve *Tet* knockdown specifically in niche cells at adult stage, *bab1-Gal4* is combined with a temperature-sensitive mutant Gal80 (Gal80$^{ts}$) controlled by the ubiquitous tubulin gene promoter (*tub-Gal80$^{ts}$*). Flies carrying *bab1-Gal4, tub-Gal80$^{ts}$ (bab1$^{ts}$)*, and *UAS-Tet-shRNA* were raised at 21 °C during development to suppress Gal4 function and consequently inhibit *Tet* shRNA expression. Subsequently, the flies were shifted to 29 °C to deactivate Gal80$^{ts}$ and activate Gal4-mediated expression of *Tet* shRNA in adulthood. In this study, we used knockdown of the firefly luciferase gene (*luc-KD*) as the control because this is a foreign gene that does not exist in the *Drosophila* genome.

To assess the effect of *Tet* knockdown in adult niche cells, two independent transgenic shRNA lines (*bab1$^{ts}$>Tet-KD1* and *bab1$^{ts}$>Tet-KD2*) were employed. Both lines were able to significantly reduce the expression of *Tet* mRNA and protein (Tet-EGFP), demonstrating their efficient knockdown capability in niche cells (Fig. EV1D–G). To examine the impact of *Tet* knockdown on cap cells, GSCs, and CBs, we used an antibody against Hu-li tai shao (Hts) to label *luc-KD* and *Tet-KD* ovaries. Hts is a marker for the spherical spectrosome in GSCs and CBs, as well as the branched fusome in cysts. By observing the direct contact between cap cells and Hts-labeled germ cells, GSCs can be reliably distinguished from CBs. Interestingly, *Tet-KD* germaria still contained approximately 5-7 cap cells, similar to the *luc-KD* ones, indicating that Tet is not essential for cap cell maintenance (Fig. 1C,D). The *Tet-KD1* and *Tet-KD2* germaria significantly decrease GSCs numbers compared to the *luc-KD* control, while the control and *Tet* knockdown germaria exhibit similar CB numbers (Fig. 1C,D). These data indicate that Tet is necessary in the niche to maintain GSC self-renewal but not for the integrity of the niche itself.

### Tet is required in the niche to activate BMP signaling for controlling GSC self-renewal

Cap cells use E-cadherin-mediated cell adhesion to anchor GSCs and also secrete BMP-like Dpp for promoting GSC self-renewal by repressing the expression of the key differentiation factor Bam (Song et al, 2004; Song et al, 2002). In the *Tet-KD1* and *Tet-KD2* germaria, E-cadherin accumulation in the GSC-niche junction remains normal compared to the *luc-KD* germaria, indicating that Tet is dispensable for E-cadherin expression in the niche (Fig. EV2A,B). Niche-produced Dpp binds to its receptors in GSCs to activate a kinase cascade leading to Mad phosphorylation (pMad) and the repression of *bam* transcription (Chen and McKearin, 2003; Song et al, 2004; Xie and Spradling, 1998). To investigate BMP signaling in GSCs, we examined the expression of two BMP signaling indicators, pMad and *bam*-GFP (a GFP reporter for *bam* transcription). In the *luc-KD* germaria, GSCs express pMad but not *bam*-GFP, while CBs and mitotic cysts express *bam*-GFP but low or no pMad (Fig. 1E–H). In contrast, GSCs in the *Tet-KD* germaria often downregulate pMad expression and upregulate *bam*-GFP expression (Fig. 1E–H).

Furthermore, through RNA FISH analysis, we observed a significant downregulation of *dpp* mRNA expression in the *Tet-KD1* and *Tet-KD2* germaria compared to the *luc-KD* (Fig. 1I,J). Interestingly, the GSC loss resulting from *Tet-KD1* and *Tet-KD2*

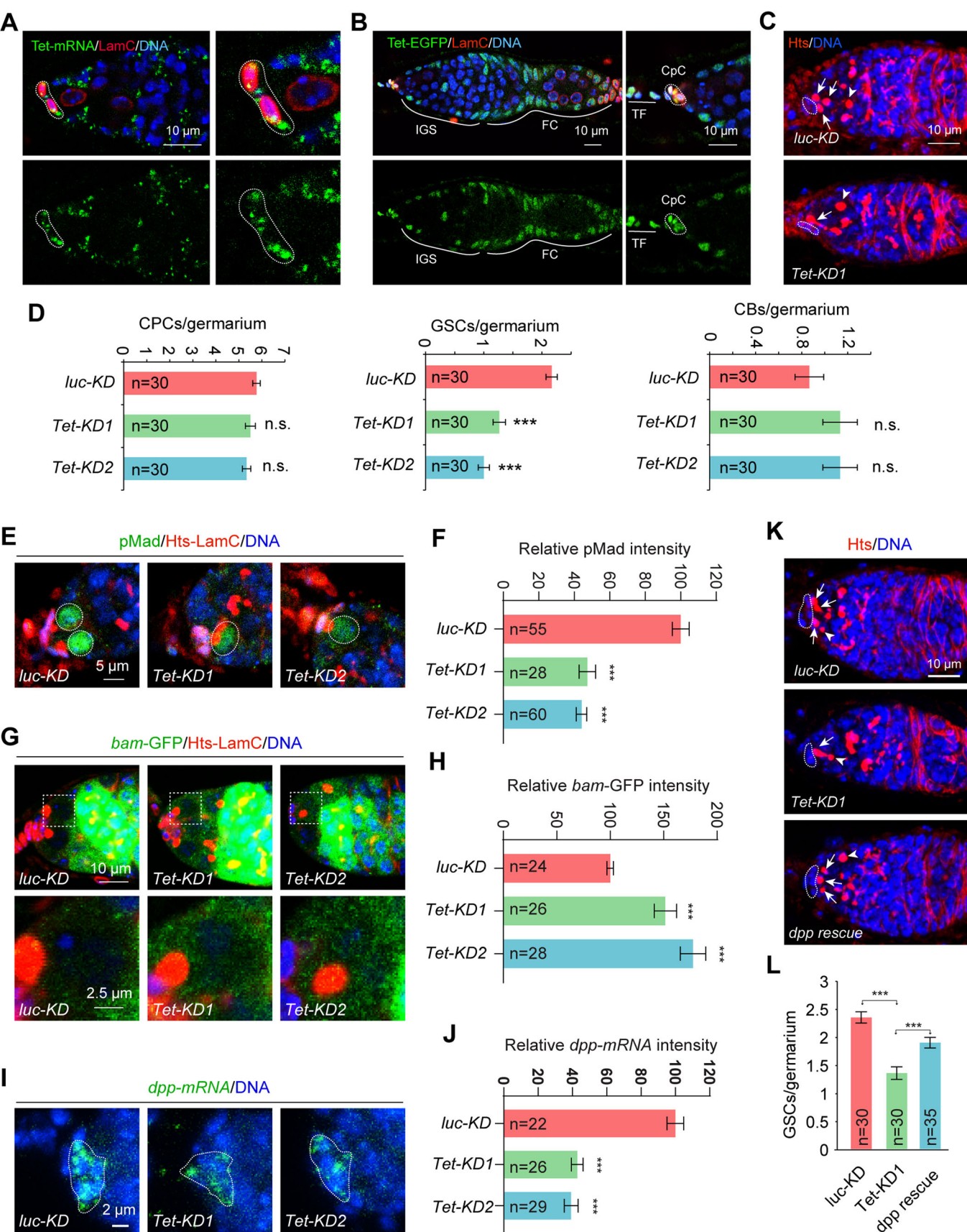

**Figure 1.  Tet is required in the niche to control GSC self-renewal by activating *dpp* expression.**

(A) RNA FISH results show that *Tet* mRNA is expressed in the niche cells and IGS cells of the germarium. Niche cells are highlighted by antibody against Lamin C (LamC), which is enriched in the nuclear membrane of terminal filament cells and cap cells. (B) Tet-EGFP is expressed in the terminal filaments, cep cells, IGS cells, and follicle cells of the germarium. (C, D) Cap cells (CPCs), GSCs and CBs are highlighted by dashed white circles, arrows, and arrowheads, respectively (the same applies to images in the remaining Figures). *bab1^ts*-driven Tet knockdown causes the GSC loss but does not affect cap cell and CB numbers. (D) Quantification results. (E, F) *bab1^ts*-driven Tet knockdown decreases pMad expression in GSC (the '*luc-KD*' image is re-displayed in Fig. 4C). (F) Quantification results. (G, H) *bab1^ts*-driven Tet knockdown increases *bam*-GFP expression in GSC. (H) Quantification results. (I, J) *bab1^ts*-driven Tet knockdown decreases *dpp* mRNA expression in cap cells. (J) Quantification results. (K, L) *bab1^ts*-driven *dpp* overexpression can restore GSC numbers compared with *Tet-KD*. (L) Quantification results. Data information: In (D, F, H, J and L), data are presented as mean ± SEM. ***P ≤ 0.001, n.s., no significance (Student's t-test). In (D, J and L), n = number of germaria. In (F and H), n = number of GSCs. Source data are available online for this figure.

can be significantly rescued by *bab1^ts*-mediated *dpp* overexpression in niche cells (Fig. 1K,L). These findings collectively reveal that Tet plays a crucial role in the niche by activating *dpp* expression and maintaining active BMP signaling in GSCs, ultimately controlling GSC self-renewal.

## Tet functions independently of its dioxygenase activity in the niche to promote GSC self-renewal

To investigate if the dioxygenase activity of Tet is required in the niche to maintain GSCs, we generated two transgenic strains: *UASz-Flag-Tet^WT* and *UASz-Flag-Tet^ED*. These strains contain RNAi-resistant wild-type (WT) and enzyme-dead (ED) *Tet* genes, respectively, under the control of the *UASz* promoter (resistant to BL62280) (Fig. 2A). Overexpression of *Tet^WT* and *Tet^ED* using *bab1^ts* did not result in any noticeable defects in the germaria, including GSC development (Fig. 2B–G). Remarkably, overexpression of *Tet^WT* fully rescued the GSC numbers, pMad expression in GSCs, and *dpp* expression in niche cells in the *Tet-KD* germaria, confirming that the observed phenotypes in the *Tet-KD* germaria were indeed caused by the depletion of *Tet* expression (Fig. 2B–G). Similarly, overexpression of *Tet^ED* also completely restored the GSC numbers, pMad expression in GSCs, and *dpp* expression in niche cells in the *Tet-KD* germaria, indicating that enzymatically inactive Tet still functions in the niche to support BMP signaling and GSC self-renewal (Fig. 2B–G). These results collectively demonstrate that Tet regulates GSC self-renewal by promoting dpp transcription independently of its enzymatic activity.

## Tet-associated Bap55 functions in the niche to control GSC self-renewal by promoting *dpp* expression

To gain mechanistic insight into Tet function in the stem cell niche, we employed Flag-Tet-mediated pulldown and liquid chromatography-mass spectrometry (LC-MS) techniques. Through this approach, we identified a total of 279 putative Tet-associated proteins in S2 cells (Fig. 3A). Interestingly, the Tet-associated proteins identified include Act5C, Bap55 and Bap111, which are components of the Brahma (Brm)-associated protein complexes (Fig. 3A). To further confirm and map the interaction between Tet and Bap55, we expressed Tet protein in three fragments: N-terminal Tet (Tet-N), mid-region Tet (Tet-M), and C-terminal Tet (Tet-C). Interestingly, only Tet-C, but not Tet-N and Tet-M, can bring down Bap55-HA (Fig. 3B,C). To further map the Tet protein region interacting with Bap55, Tet-C was then expressed in four independent proteins fragments, Tet-CF1 to Tet-CF4, with Bap55-HA in S2 cells. Both Tet-CF1 and Tet-CF4, but not Tet-CF2

and Tet-CF3, can bring down Bap55-HA (Fig. 3B,C). Taken together, these results indicate that Tet interacts with Bap55 through it two C-terminal regions, Tet-CF1 and Tet-CF4.

To assess the role of Bap55 and Bap111 in controlling GSC self-renewal within the niche, we proceeded to knock down the expression of them in adult niche cells and examined the numbers of cap cells, GSCs, and CBs. Quantitative real-time PCR (qPCR) analysis confirmed that RNAi against *Bap55* and *Bap111* can significantly reduce their mRNAs, respectively (Appendix Fig. S1A,B). Interestingly, both *Bap55* and *Bap111* knockdowns significantly decreased the number of GSCs compared to *luc-KD*, while having no noticeable effect on CBs and cap cells (Fig. 3D,E). This finding indicates that Bap55 and Bap111 are crucial in the niche for maintaining GSC self-renewal. Similar to *Tet* knockdown, both *Bap55* and *Bap111* knockdowns in niche cells led to a decrease in pMad expression in GSCs and a reduction in *dpp* mRNA expression in niche cells (Fig. 3F–I). Taken together, these results suggest that Tet-associated Bap55 and Bap111 function in the niche to control GSC self-renewal by activating *dpp* expression and maintaining active BMP signaling.

## The PBAP complex functions in the niche to activate *dpp* expression and control GSC self-renewal

The Brm protein is involved in forming two distinct protein complexes, Brahma associated proteins (BAP) complex and Polybromo Brahma associated protein (PBAP) complex, which share several components such as Act5C, Moira, SNR1, BRM, Bap55, BAP60, and Bap111 (Hong and Choi, 2016) (Fig. EV3A). Here, we employed a Bap111-GFP in which GFP is inserted in-frame within the genomic locus of a BAC transgene. This results in the presence of both the two endogenous Bap111 copies and the BAC's Babp111-GFP. We observed that the expression of Bap111-GFP remains unaltered in *Tet-KD* niche cells (Fig. EV3B,C), suggesting that Tet may not play a significant role in maintaining the expression level of Bap111-containing BAP and PBAP complexes within the niche. The BAP complex has one unique component Osa, whereas the PBAP complex contain three specific components, Bap170, Bap180, and SAYP (Hong and Choi, 2016) (Fig. EV3A). To determine which complex is involved in controlling *dpp* expression, we sought to knock down the expression of *brm*, *Bap170*, *Bap180*, and *osa* in adult niche cells. qPCR analysis confirmed that RNAi strains against these genes can significantly reduce the expression the *brm*, *Bap170*, *Bap180*, and *osa* mRNAs, respectively (Appendix Fig. S1C–F). Subsequently, we observed that knockdowns of *brm*, *Bap170*, *Bap180*, and *osa* led to a significant decrease in the number of GSCs compared to *luc-KD*

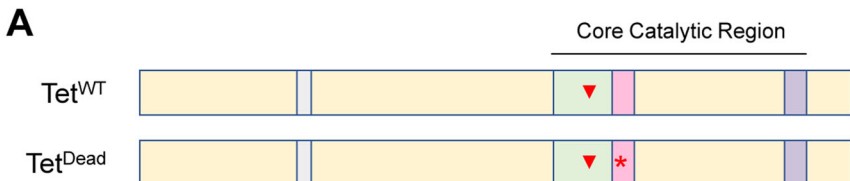

**A**

Core Catalytic Region

Tet^WT

Tet^Dead

▼ RNAi resistant: WSMYY: TGGTCGATGTACTAC to TGATCCATGTATTAT

\* Enzyme Dead: HSHRDL to HSARAL

**B** Hts-LamC/DNA

*luc-KD* — *Tet-KD1*

10 μm

*Tet^WT-OE* — *Tet^ED-OE*

*Tet^WT-Res* — *Tet^ED-Res*

**D** pMad/Hts-LamC/DNA

*luc-KD* — *Tet-KD1*

5 μm

*Tet^WT-OE* — *Tet^ED-OE*

*Tet^WT-Res* — *Tet^ED-Res*

**F** *dpp-mRNA*/DNA

*luc-KD* — *Tet-KD1*

5 μm

*Tet^WT-OE* — *Tet^ED-OE*

*Tet^WT-Res* — *Tet^ED-Res*

**C**

GSCs/germarium (n=108, n=85, n=60, n=57, n=59, n=62 for luc-KD, Tet-KD1, Tet^WT-OE, Tet^ED-OE, Tet^WT-Res, Tet^ED-Res)

**E**

Relative pMad intensity (n=19, n=17, n=17, n=17, n=17, n=36)

**G**

Relative *dpp* intensity (n=21, n=19, n=13, n=13, n=13, n=13)

control (Fig. 4A,B). These findings indicate that both the BAP and PBAP complexes are necessary in the niche for controlling GSC self-renewal.

Then, we determined if both BAP and PBAP complexes are also required in the niche to activate *dpp* expression and maintain active BMP signaling in GSCs. Consistent with *Bap55-KD* and *Bap111-KD*, we found that *brm-KD* can also decrease *dpp* expression in niche cells and pMad expression in GSCs (Fig. 4C–F). Interestingly, *Bap170-KD* and *Bap180-KD* significantly reduced *dpp* expression in the niche and attenuated pMad expression in GSCs, but *osa-KD* had no obvious impact on *dpp* and pMad expression, indicating that the PBAP complex, but not the BAP complex, is required in the niche to control

**Figure 2. Tet functions in the niche independent of its enzymatic activity.**

(A) Domains of the Tet protein and the design of wild-type Tet (Tet*WT*, RNAi-resistant) and enzyme-dead Tet (Tet*ED*, RNAi-resistant) transgenic flies. Codons in the shRNA (BDSC #62280) target region are replaced with synonymous codons that do not change the amino acids. For the Tet*ED*, HSHRDL in core catalytic region is mutated to HSARAL. (B, C) Tet*WT* and Tet*ED* overexpression (OE) do not affect the GSC number. Both Tet*WT*-OE and Tet*ED*-OE can significantly rescue the GSC loss caused by Tet-KD. (C) Quantification results. (D, E) Tet*WT*-OE and Tet*ED*-OE do not change pMad expression in GSCs. Both Tet*WT*-OE and Tet*ED*-OE can significantly rescue the decreased pMad expression caused by Tet-KD. (E) Quantification results (n = GSC numbers). (F, G) Tet*WT*-OE and Tet*ED*-OE have no obvious effect on *dpp* expression in cap cells. Both Tet*WT*-OE and Tet*ED*-OE can significantly rescue the deceased *dpp* mRNA expression caused by Tet-KD. (G) Quantification results. Data information: In (C, E and G), data are presented as mean ± SEM. ***$P \leq 0.001$, n.s., no significance (Student's t-test). In (C and G), $n$ = number of germaria. In (E), $n$ = number of GSCs. Source data are available online for this figure.

GSC self-renewal by maintaining BMP signaling (Fig. 4C–F). In addition, *osa-KD*, but not *brm-KD*, *Bap170-KD*, and *Bap180-KD*, significantly decreased E-cadherin accumulation at the GSC-niche junction compared to the *luc-KD* (Fig. EV4A,B). These results indicate that the PBAP complex is required in the niche to maintain active BMP signaling and GSC self-renewal, which has prompted us to hypothesize that Tet works with the PBAP complex in the niche to control GSC self-renewal by promoting BMP signaling in GSCs (Fig. EV4C,D). The knockdowns of core subunits, including *brm*, *Bap55,* and *Bap111*, did not have any effect on the expression of E-cadherin, which suggests that neither the PBAP nor the BAP complex is required for maintaining the expression of E-cadherin. However, it is worth noting that the observed reduction in E-cadherin expression in *osa-KD* germaria could potentially be attributed to the function of Osa independent of BAP complex (Fig. EV4C,D).

## Tet functions as a protein scaffold to recruit the PBAP complex to Stat92E to activate *dpp* expression in the niche and promote GSC self-renewal

Previous studies have shown that Jak-Stat92E signaling in cap cells activates the expression of *dpp* for controlling GSC self-renewal (Lopez-Onieva et al, 2008; Wang et al, 2008). Consistently, we observed that *bab1ts*-driven *Stat92E* knockdown in niche cells leads to the reduced expression of *dpp* mRNA in the niche and thus premature GSC loss, further confirming that Stat92E functions in the niche to control GSC self-renewal by maintaining BMP Jak-Stat (Appendix Fig. S2A–C). The similar phenotypes observed in *Tet-KD* and *Stat92E-KD* led us to hypothesize an association between Tet and Stat92E. To investigate this, we conducted co-IP experiments in S2 cells by co-expressing Flag-tagged Tet-N, Tet-M, and Tet-C with Stat92E-HA. Flag-Tet-C, but not Flag-Tet-N or Flag-Tet-M, can pull down Stat92E-HA in S2 cells, indicating that Stat92E is associated with the C-terminus of Tet (Fig. 5A). In addition, Tet-CF1 and Tet-CF4, but not Tet-CF2 and Tet-CF3, can pull down Stat92E-HA in S2 cells, indicating that Stat92E is associated with the two C-terminal regions of Tet (Fig. 5B). These results indicate that Tet can interact with Stat92E using its two C-terminal regions.

Tet interacts with both Stat92E and Bap55 using the same two C-terminal regions of Tet. Based on this observation, we propose a working model in which Tet acts as a protein scaffold to recruit the PBAP complex to Stat92E, thereby activating *dpp* expression in the niche. To test this model, we conducted in vitro experiments using purified Tet-C, Stat92E, and Bap55 proteins. Myc-Tet-C and HA-Stat92E proteins are purified from bacteria, whereas Strep-Bap55 protein is purified from insect cells (Fig. 5C). Excitingly, both Bap55 and Stat92E can directly interact with Tet in vitro, while Bap55 and Stat92E cannot bring down each other (Fig. 5D–F). Only

in the presence of Tet-C proteins, Stat92E can pull down Bap55 (Fig. 5F). These findings provide evidence that Tet can function as a scaffold protein, bringing together Bap55 and Stat92E.

To further investigate our hypothesis, we constructed a *UASz-Flag-Bap55-Stat92E* transgenic strain expressing a Bap55-Stat92E fusion protein to test if the Bap55-Stat92E fusion protein can bypass the requirement of Tet in the niche to control GSC self-renewal. Interestingly, niche-specific overexpression of Bap55 alone or activated Stat92E alone cannot rescue the mutant phenotypes caused by *Tet-KD* in the niche, including the GSC loss, the decreased pMad expression in GSCs, and the reduced *dpp* expression in niche cells, indicating that increasing Bap55 or Jak-Stat signaling cannot bypass the requirement of Tet in niche cells to maintain BMP signaling and GSC self-renewal (Fig. 5G–L). Excitingly, niche-specific overexpression of the Bap55-Stat92E fusion protein can fully rescue all the phenotypes caused by niche-specific *Tet* knockdown, including the GSC loss, the attenuated pMad expression in GSCs, and the downregulated *dpp* expression in niche cells (Fig. 5G–L).

The *dpp* gene has four predicted promoters, namely *P1-P4* (Dreos et al, 2015; Meylan et al, 2020) (Fig. 6A). Furthermore, a recent study has discovered the significance of an enhancer called *dpp2.0*, which is occupied by Engrailed and Nejire, in promoting *dpp* expression in niche cells (Luo et al, 2017) (Fig. 6A). We subsequently designed and conducted chromatin immunoprecipitation (ChIP) experiments, and the results showed that Tet-EGFP, Bap55-HA (*bab1ts>Bap55-HA*), and Stat92E-HA (*bab1ts>Stat92E-HA*) were able to pull down the *dpp2.0* fragment, rather than the four promoters (Fig. 6B–D). Interestingly, after knockdown of *Tet*, both Bap55-HA and Stat92E-HA exhibited a significant decrease in the enrichment effect on *dpp2.0* (Fig. 6E,F). Taken together, these results reveal that Tet functions as a protein scaffold to recruit the PBAP complex to Stat92E to activate *dpp* expression in the niche for maintaining BMP signaling and GSC self-renewal.

## Human TET3 can functionally replace *Drosophila* Tet to maintain GSC self-renewal

To determine if Tet function is conserved from *Drosophila* to humans, we used a *UAS-hTET3* (human TET3) transgene to overexpress Tet in *luc-KD* control and *Tet-KD* niches in combination with *bab1ts*. Excitingly, niche-specific overexpression of hTET3 can fully rescue all the mutant phenotypes caused by *Tet* knockdown in the niche, including the GSC loss, the decreased pMad expression in GSCs and the downregulated *dpp* expression in niche cells (Fig. 7A–F). These results demonstrate that human TET3 can functionally replace *Drosophila* Tet in the niche to maintain BMP signaling and control GSC self-renewal.

                                                                   

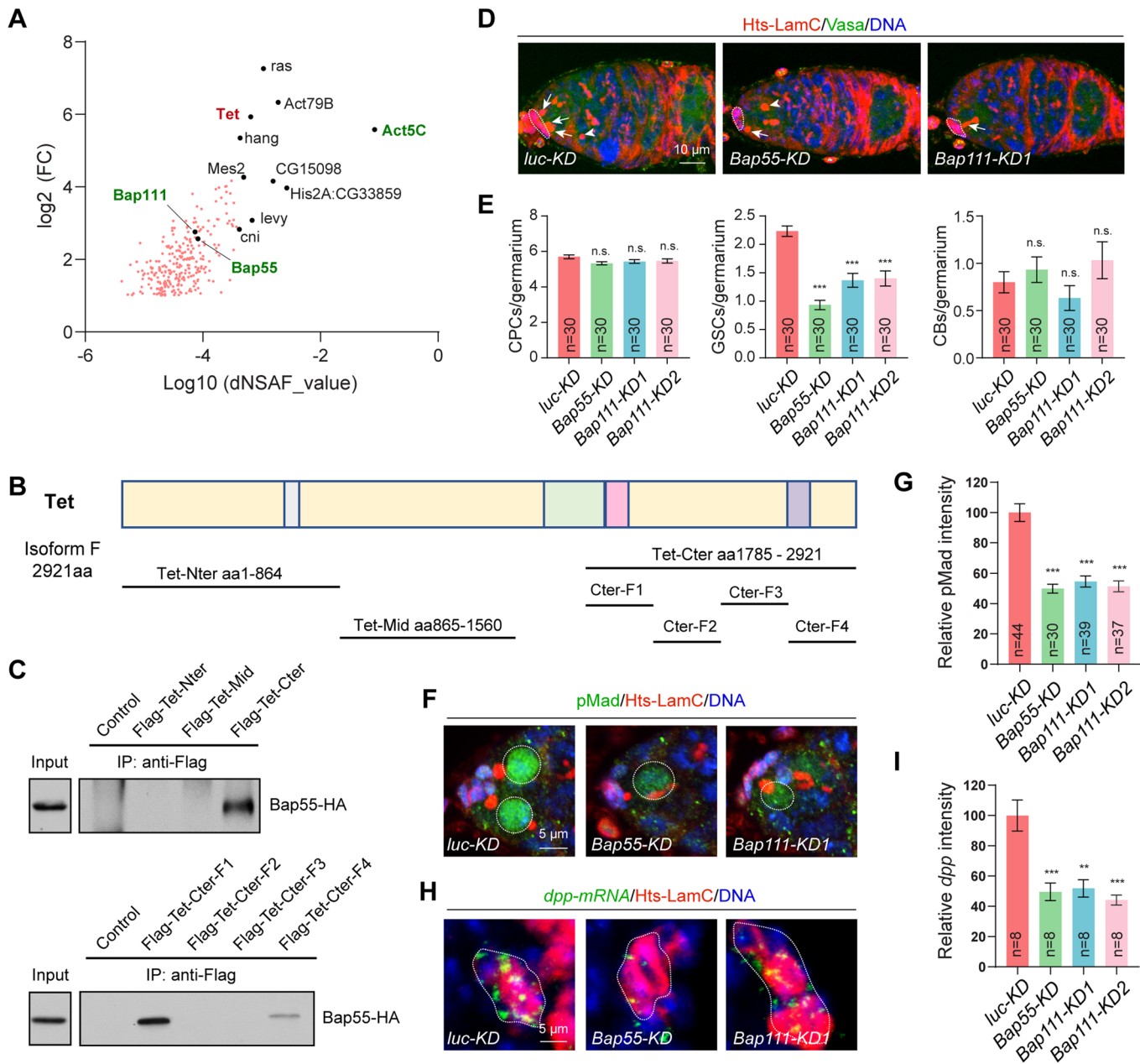

**Figure 3. Tet physically associates with Bap55.**

(A) Diagram shows Flag-Tet associated proteins identified by co-IP and mass spectrometry, including PBAP components, Bap55, Bap111, and Act5C. (B, C) Co-IP experiments show that Flag-tagged Tet C-terminal fragment (Flag-Tet-C) can bring down Bap55-HA in S2 cells. Moreover, Flag-Tet-CF1 and Flag-Tet-CF4 can bring down Bap55-HA in S2 cells. (D, E) *bab1ᵗˢ*-driven *Bap55* and *Bap111* knockdowns cause the GSC loss but do not affect cap cell and CB numbers. (E) Quantification results. (F–I) *bab1ᵗˢ*-driven *Bap55* and *Bap111* knockdowns reduce pMad (F) expression in GSCs and decrease *dpp* mRNA (H) expression in cap cells. (G, I) Quantification results. Data information: In (E, G and I), data are presented as mean ± SEM. **P ≤ 0.01, ***P ≤ 0.001, n.s., no significance (Student's t-test). In (E and I), n = number of germaria. In (G), n = number of GSCs. Source data are available online for this figure.

## Discussion

Although Tet family proteins have been extensively studied for their role in DNA methylation and epigenetics, their non-dioxygenase activity functions are not well understood. In the adult *Drosophila* ovary, Tet has been suggested to function intrinsically for promoting GSC progeny differentiation, which

has been disputed by another independent study (Wang et al, 2018; Zhang et al, 2015). In our study, we discovered a novel function of Tet in the niche: maintaining GSC self-renewal by promoting BMP signaling independently of its enzymatic activity. Tet acts as a protein scaffold, interacting directly with Bap55 and Stat92E, to recruit the PBAP complex to Stat92E. The Bap55-Stat92E fusion protein can compensate for the requirement of Tet in the niche to

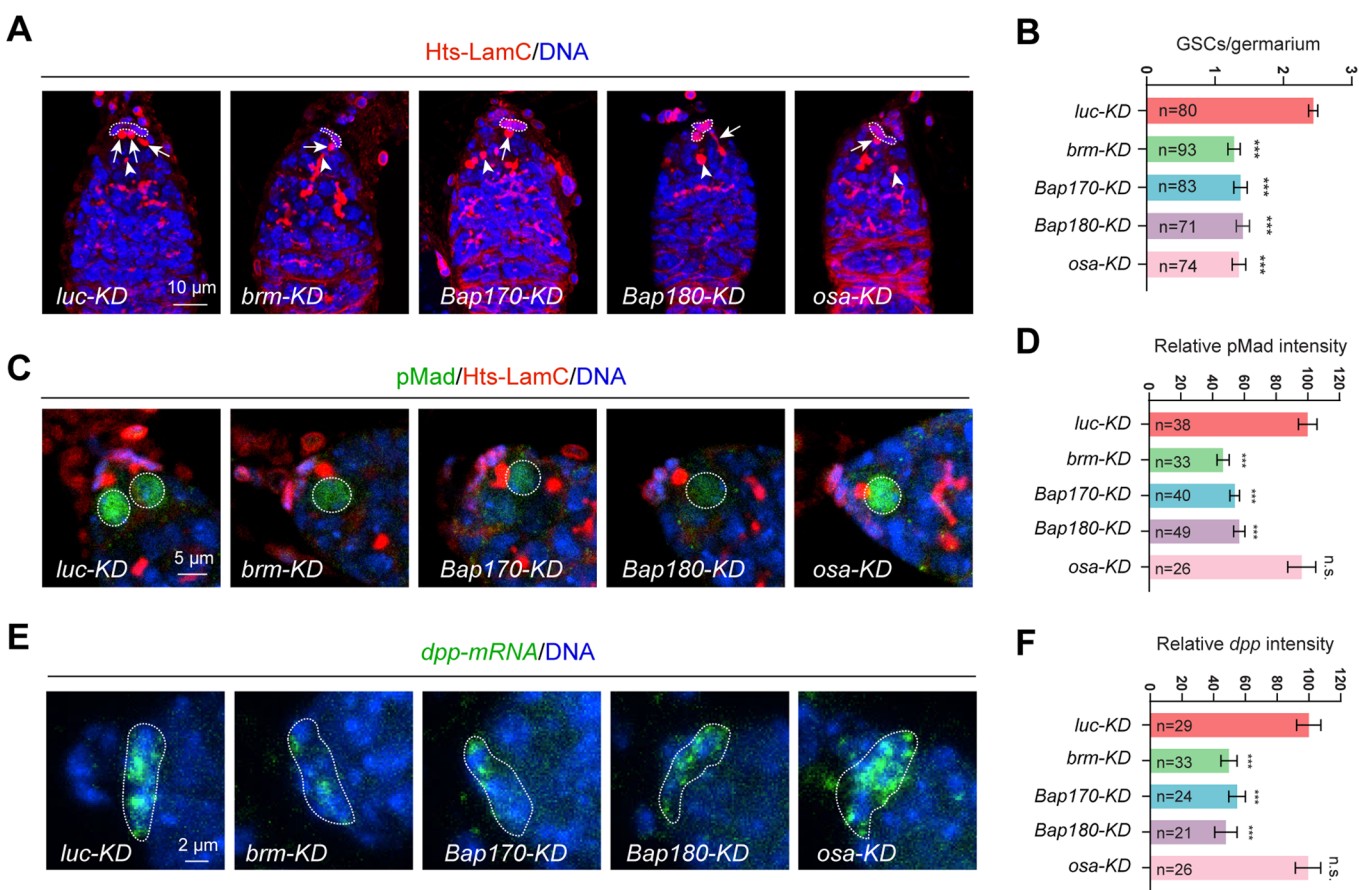

**Figure 4. The PBAP complex is required in cap cells to promote *dpp* expression.**

(A, B) *bab1ts*-driven *brm-KD*, *Bap170-KD*, *Bap180-KD*, and *osa-KD* decrease GSC number. (B) Quantification results. (C, D) *bab1ts*-driven *brm-KD*, *Bap170-KD*, and *Bap180 KD* decrease pMad expression in GSCs, but *osa-KD* has no obvious effect on pMad expression in GSCs. (D) Quantification results. (E, F) *bab1ts*-driven *brm-KD*, *Bap170-KD*, and *Bap180-KD* decrease *dpp* expression in cap cells, but *osa-KD* has no obvious effect on *dpp* expression in cap cells. (F) Quantification results. Data information: In (B, D and F), data are presented as mean ± SEM. ***P ≤ 0.001, n.s., no significance (Student's t-test). In (B and F), n = number of germaria. In (D), n = number of GSCs. Source data are available online for this figure.

sustain BMP signaling and GSC self-renewal. Furthermore, human TET3 can replace *Drosophila* Tet in the niche and support BMP signaling and GSC self-renewal. Based on our findings, we propose that Tet operates independently of its enzymatic activity, facilitating the interaction between the PBAP complex and Stat92E, resulting in the activation of *dpp* expression in the niche. This activation is vital for maintaining GSC self-renewal (Fig. 7G).

## Tet functions in the niche to control GSC self-renewal by maintaining active BMP signaling

Tet was initially proposed to function as a DNA 6 mA demethylase and be expressed in early GSC progeny to promote their differentiation in the adult *Drosophila* ovary (Zhang et al, 2015). However, another study uses marked *Tet* mutant GSC clones to show that Tet is dispensable for early GSC progeny differentiation (Wang et al, 2018). In addition, Tet is expressed in somatic cells, but not in germ cells, of the adult ovary using a Tet-GFP reporter line, in which GFP is inserted in the middle of the Tet protein (Wang et al, 2018). Due to the complicated *Tet* splicing patterns, the GFP insertion strain might not reflect the expression patterns of

all *Tet* transcripts. This study has used two independent approaches, mRNA FISH and C-terminal knock-in Tet-EGFP, to show that *Tet* mRNA and Tet protein exhibit identical expression patterns in all the somatic cells of the ovary, including cap cells, but not in germ cells. Our C-terminal knock-in Tet-EGFP strain should label all the protein isoforms generated by alternative splicing. Therefore, our results help rule out the possibility that Tet is expressed in GSC progeny to control their differentiation in the adult *Drosophila* ovary.

In this study, we demonstrate that Tet is required in the niche to control GSC self-renewal by maintaining active BMP signaling. Cap cell-expressed Dpp/BMP has been shown to directly activate BMP signaling in GSCs and promote GSC self-renewal by preventing the expression of differentiation-promoting genes, including *bam* (Chen and McKearin, 2003; Song et al, 2004; Xie and Spradling, 1998, 2000). We used the niche-enriched *bab1-Gal4* driver and two independent *UAS-shRNA* lines to knock down *Tet* expression specifically in the niche to show that Tet is required in the niche to control GSC self-renewal. In addition, niche-specific *Tet* knockdown downregulates *dpp* expression in cap cells and pMad expression in GSCs and upregulates *bam*-GFP expression in GSCs,

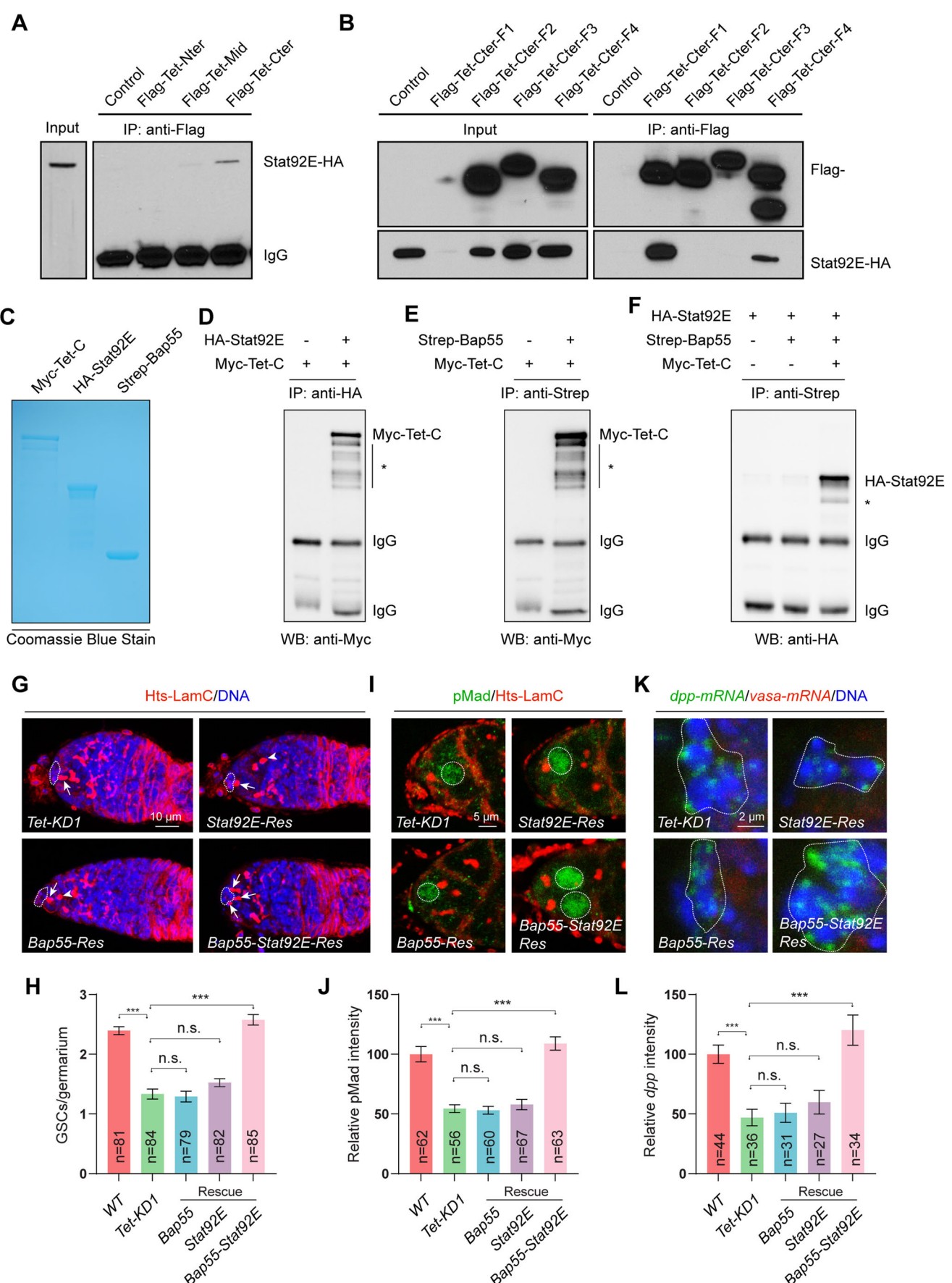

**Figure 5.  Tet recruits the PBAP complex to Stat92E to activate *dpp* expression by directly interacting with Bap55 and Stat92E.**

(A, B) Co-IP experiments show that Flag-tagged Tet C-terminus (Flag-Tet-C) can bring down Stat92E-HA in S2 cells. Moreover, Flag-Tet-CF1 and Flag-Tet-CF4 (two smaller regions in Tet-C) can bring down Stat92E-HA in S2 cells. (C) Coomassie blue staining on purified Myc-Tet-C and HA-Stat92E from bacteria, and Strep-Bap55 from insect cells. (D–F) In vitro interaction assay shows that Myc-Tet-C directly interacts with HA-Stat92E (D) and Strep-Bap55 (E). Strep-Bap55 brings down HA-Stat92E only in the presence of Myc-Tet-C, but not Strep-Bap55 alone (F). (*Indicates that these bands may be a result of protein instability and degradation) (G, H) *bab1ᵗˢ*-driven Bap55-Stat92E overexpression can significantly rescue the GSC loss caused by *Tet-KD*, but overexpression of Bap55 or Stat92E alone cannot. (H) Quantification results. (I, J) *bab1ᵗˢ*-driven Bap55-Stat92E overexpression can significantly rescue the decreased pMad expression caused by *Tet-KD*, but overexpression of Bap55 or Stat92E alone cannot. (J) Quantification results. (K, L) *bab1ᵗˢ*-driven Bap55-Stat92E overexpression can significantly rescue the reduced *dpp* mRNA expression caused by *Tet-KD*, but overexpression of Bap55 or Stat92E alone cannot. (L) Quantification results. Data information: In (H, J, and L), data are presented as mean ± SEM. ***$P \leq 0.001$, n.s., no significance (Student's t-test). In (H and L), n = number of germaria. In (J), n = number of GSCs. Source data are available online for this figure.

indicating that Tet is required in the niche to maintain BMP signaling. Overexpressing *dpp* in the niche can significantly rescue the GSCs loss caused by *Tet* knockdown in the niche, indicating that one of the major functions of Tet in the niche is to maintain Dpp expression. Therefore, our findings demonstrate that Tet functions in the niche to control GSC self-renewal by promoting Dpp/BMP signaling. This has uncovered a novel function of Tet in the niche for controlling stem cell self-renewal, representing the first demonstration of such function in the stem cell niche.

Interestingly, we did not observe a reduction in CBs in *Tet-KD* germaria, despite the decrease in GSC numbers. The reduction of GSCs caused by *Tet-KD* in our study is not attributed to decreased adhesion between GSCs and the niche. Instead, it is a result of weakened BMP signaling in GSCs, leading to excessive differentiation. We speculate that the reasons for the unchanged CBs may involve factors such as slow CB differentiation or division.

## Tet functions as a protein scaffold independently of its enzymatic activity in the niche to recruit the PBAP complex to Stat92E to control BMP signaling and GSC self-renewal

TET proteins have dioxygenase activity-dependent and -independent functions in the regulation of various developmental processes (Wu and Zhang, 2017; Yang et al, 2020). The dioxygenase activity-dependent functions of TET proteins are critical for epigenetic regulation in DNA and RNA and involved in transcriptional regulation, mRNA stability and translation (Lan et al, 2020; Wu and Zhang, 2017; Yang et al, 2020). However, the enzymatic activity-independent functions of TET proteins have recently been revealed as well. In the early post-implantation mouse embryo where TET1, but not TET2 and TET3, is expressed, mutations in the catalytic activity of *Tet1* have little or no defects on embryonic development (Dawlaty et al, 2013; Kang et al, 2015; Zhang et al, 2013), but a *Tet1* null mutation causes severe embryonic defects (Khoueiry et al, 2017), indicating that TET1 has an enzymatic activity-independent function in regulating embryonic development. Similarly, TET1 also represses gene expression in adipocytes in a DNA demethylation-independent manner (Damal Villivalam et al, 2020). Moreover, TET2 directly interacts with O-linked β-N-acetylglucosamine transferase to promote transcription independent of its enzymatic function (Chen et al, 2013). Furthermore, TET1 associates with SIN3A to silence endogenous retroviral elements independently of DNA methylation (Stolz et al, 2022).

In this study, we used a combination of genetics and biochemistry to demonstrate that Tet functions independently of

its dioxygenase activity in the niche to control GSC self-renewal by regulating BMP signaling. First, we demonstrate that the dioxygenase-inactive Tet can fully rescue all the mutant phenotypes caused by Tet knockdown in the niche in the *Drosophila* ovary like the wild-type Tet, including the GSC loss, the downregulated *dpp* expression in cap cells and the decreased pMad expression in GSCs, demonstrating that Tet functions independently of its dioxygenase activity in the niche to control GSC self-renewal by promoting BMP signaling. Second, we demonstrate biochemically that Tet helps recruit the PBAP complex to Stat92E by directly interacting with PBAP component Bap55 and Stat92E, which do not interact. BAP components, Act5C, Bap55, and Bap111, were identified as potential Tet-associated proteins. In addition, Tet was shown to directly binds to Bap55 and Stat92E to form a tertiary protein complex in vitro. Our findings could also help explain the genetic interaction between Brm and Stat92E in border cells to control their migration in the *Drosophila* ovary (Saadin and Starz-Gaiano, 2016). Third, we demonstrated genetically that the PBAP complex, but not the BAP complex, functions in the niche to control GSC self-renewal by promoting BMP signaling. Knocking down PBAP-specific components, but not BAP-specific component Osa, in the niche causes the GSC loss by disrupting BMP signaling. Finally, we showed genetically that overexpressing the Bap55-Stat92E fusion protein, but not Bap55 or Stat92E alone, can bypass the requirement of Tet in the niche for maintaining BMP signaling and GSC self-renewal, indicating that Tet functions as a bridge to bring the PBAP complex to Stat92E. Jak-Stat92E signaling has previously been shown to activate Dpp expression in cap cells and maintain active BMP signaling in GSCs, thereby controlling GSC self-renewal (Lopez-Onieva et al, 2008; Wang et al, 2008).

In our in vitro protein interaction assays, the binding of Tet and Stat92E was independent of Stat92E phosphorylation, as Stat92E is expressed in *E. coli*. Therefore, we speculate that Stat92E phosphorylation is not essential for the recruitment of PABP and Stat92E by Tet as a scaffold protein. However, it is important to note that in the in vivo environment, Stat92E should be phosphorylated by activated JAK signaling before it can enter the nucleus (Herrera and Bach, 2019). In contrast, Tet and the PABP complex primarily localize in the nucleus. While Stat92E phosphorylation is not necessary for its binding with Tet/PBAP, it plays a crucial role in regulating *dpp* expression, which is consistent with the critical role of JAK/Stat signaling in controlling *dpp* expression in *Drosophila* ovarian niche cells (Lopez-Onieva et al, 2008; Wang et al, 2008). Moreover, we performed ChIP assays to show that the Tet, Bap55 and Stat92E co-occupy the *dpp2.0* enhancer in niche cells in the *Drosophila* ovary. Taken together, Tet functions independently of its enzymatic activity as a scaffold to recruit the

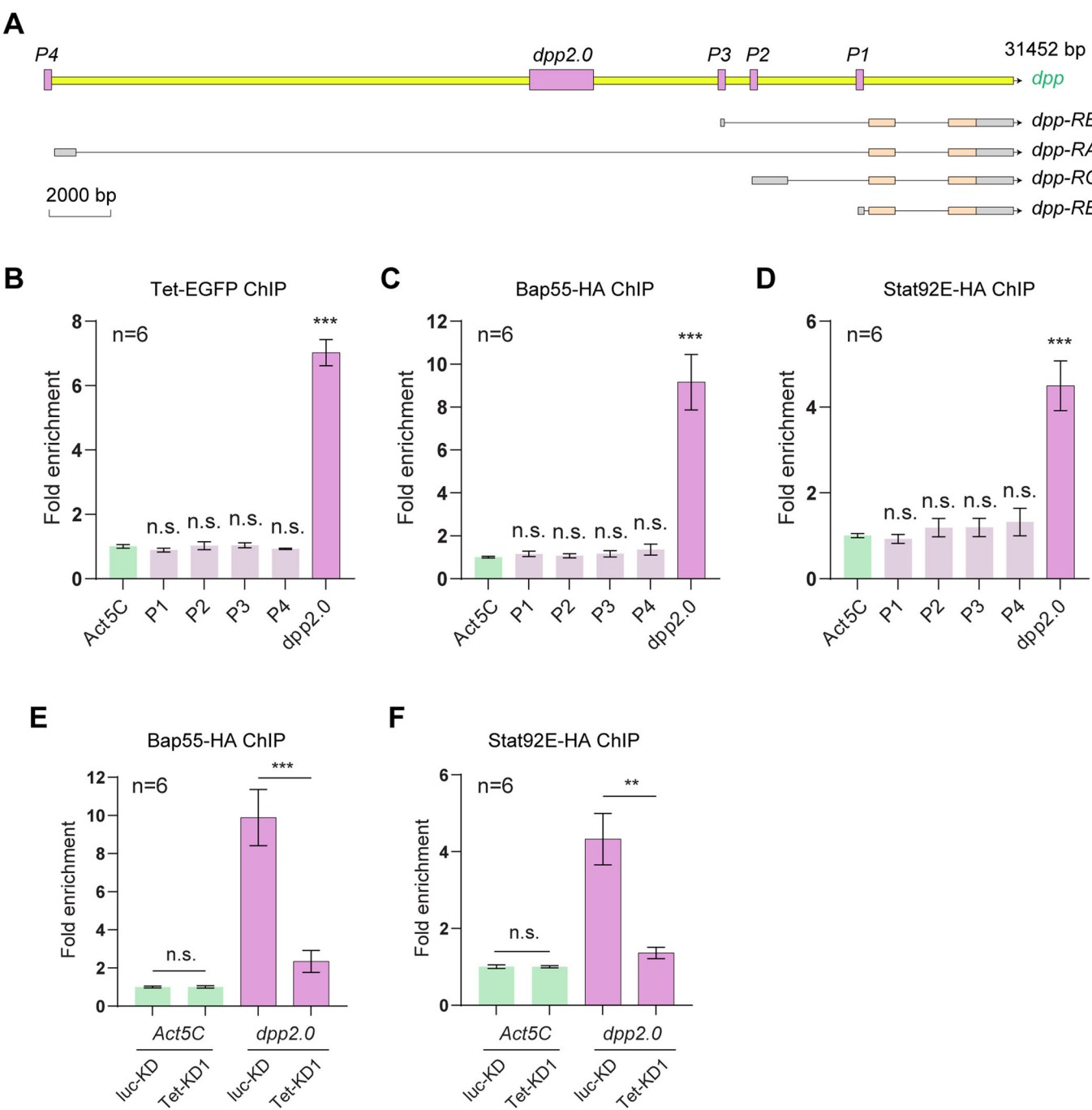

**Figure 6.   Tet, Bap55, and Stat92E bind to the *dpp2.0* enhancer.**

(A) A schematic diagram depicts the structure and transcripts, promoters (*P1-P4*) and enhancer (*dpp2.0*) of *dpp*. (B–D) ChIP results display the fold enrichment of Tet-EGFP (B), Bap55-HA (C), and Stat92E-HA (D) on the regions of *P1*, *P2*, *P3*, *P4*, and *dpp2.0*. (E, F) ChIP results reveal that the fold enrichments of Bap55-HA (E) and Stat92E-HA (F) on *dpp2.0* were significantly reduced upon *Tet-KD*. Data information: In (B–F), data are presented as mean ± SEM. **$P ≤ 0.01$, ***$P ≤ 0.001$, n.s., no significance (Student's t-test). In (B–F), $n$ = number of technical replicates. Source data are available online for this figure.

PBAP complex to Stat92E in the niche for maintaining BMP signaling and GSC self-renewal.

However, it is still unclear how Tet/Stat92E/PBAP collaborates with Engrailed/Nejire (Luo et al, 2017) to control *dpp* expression in niche cells. In future studies, it would be interesting to explore the collaborative mechanism of these transcriptional regulatory factors

in controlling gene expression at the same enhancer. Similar phenomena have been observed in other organisms, such as human embryonic stem cells, where core transcription factors OCT4, SOX2, and NANOG co-occupy a significant portion of their target genes (Boyer et al, 2005). Taken together, this study provides new insight into how Jak-Stat92E signaling activates *dpp* expression in

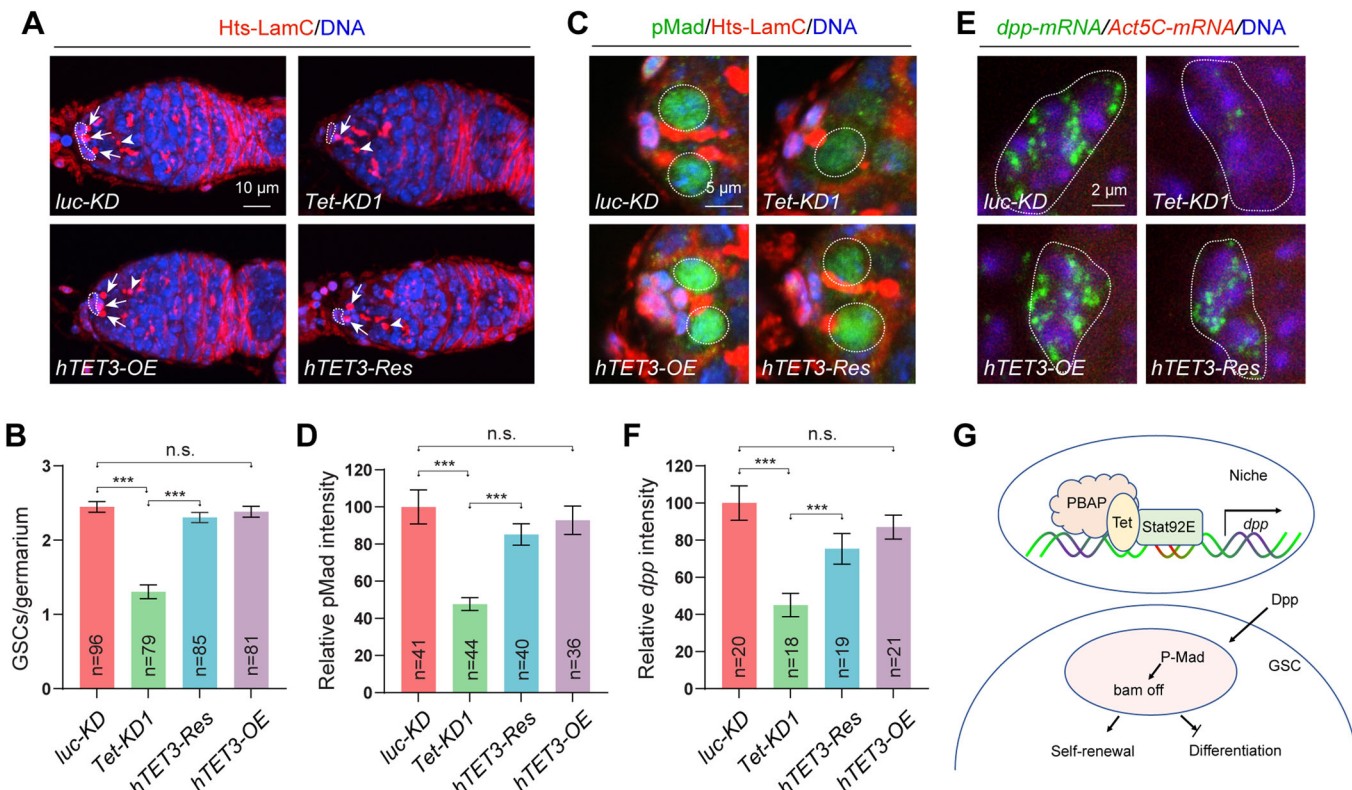

**Figure 7. Human TET3 can functionally replace *Drosophila* Tet in the niche to maintain BMP signaling and GSC self-renewal.**

(A, B) *bab1^{ts}*-driven *hTET3-OE* has no obvious impact on the GSC number in the wild-type germarium but can significantly rescue the GSC loss caused by *Tet-KD*. (B) Quantification results. (C, D) *hTET3-OE* had no obvious effect on pMad expression in wild-type GSCs but it can significantly rescue the decreased pMad expression caused by *Tet-KD*. (D) Quantification results. (E, F) *hTET3-OE* had no significant effect on *dpp* expression in wild-type cap cells but it can significantly rescue the reduced *dpp* mRNA expression in the *Tet-KD* cap cells. (F) Quantification results. (G) Schematic diagram explaining how Tet bridges the PBAP complex and Stat92E independent of its enzymatic activity to activate *dpp* expression in the niche, thus promoting GSC self-renewal. Data information: In (B, D and F), data are presented as mean ± SEM. ***$P \leq 0.001$, n.s., no significance (Student's t-test). In (B and F), $n$ = number of germaria. In (D), $n$ = number of GSCs. Source data are available online for this figure.

the niche, thereby maintaining BMP signaling in GSCs and controlling GSC self-renewal as well as how Tet functions independently of its enzymatic activity in the niche to control GSC self-renewal. Since human TET3 functionally replaces *Drosophila* Tet to activate *dpp* transcription and maintain GSC self-renewal, such novel dioxygenase-independent TET function likely works in mammals, including humans in regulating some biological processes.

# Methods

## *Drosophila* strains, culture, and treatments

The following *Drosophila* stocks used in this study are described in FlyBase, unless specified: *bab1-Gal4, bam-GFP, UAS-dpp, tub-Gal80^{ts}, tub-gal4* (BL86328), *luc-KD* (BL31603), *Tet-KD1* (BL62280), *Bap55-KD1* (BL31708), *Bap111-KD1* (BL26218), *Bap111-KD2* (BL35242), *brm-KD* (BL31712), *Bap170-KD* (BL26308), *Bap180-KD* (BL32840), *osa-KD* (BL38285), *Stat92E-KD1* (BL31318), *Stat92E-KD2* (BL33637), *UAS-Bap55* (BL34488), *Bap111-EGFP* (BL64799), *UAS-Bap55-HA* (FlyORF: F004152), *UAS-Stat92E-HA* (FlyORF: F001027) constitutively active *UAS-Stat92E^{ΔNΔC}* (Ekas et al, 2010), *UAS-hTET3* as a gift

from Dr. Shirinian (Frey et al, 2022). Flies were reared and crossed at 21 °C on standard cornmeal/molasses/agar media. For maximum RNAi-mediated knockdown, newly eclosed flies were cultured at 29 °C for 2 weeks to allow for knockdown and phenotype development before conducting dissection and immunostaining, unless otherwise specified.

## Immunofluorescence staining

Immunostaining was performed according to our previously published procedures (Mao et al, 2019; Tu et al, 2021; Tu et al, 2020; Zou et al, 2020). The following antibodies were used in this study: mouse monoclonal anti-Hts antibody (1:50, DSHB), mouse monoclonal anti-Lamin C (1:5, DSHB), rabbit polyclonal anti-β-galactosidase antibody (1:100, #08559761, MP Biomedical), rat polyclonal anti-Vasa antibody (1:50, DSHB), rabbit monoclonal anti-pMad antibody (1:500, ab52903), goat polyclonal anti-GFP antibody (1:500, Rockland, #600-101-215) and chicken polyclonal anti-GFP antibody (1:200, Invitrogen, #A10262). For rabbit polyclonal anti-Aub, a peptide (MHKSEGDPRGSVRGC) from *Drosophila* Aub was synthesized and injected into the rabbit by GeneScript USA Inc., and the Protein A/G purified IgG was used for staining (1:1000).

## Hybrid chain reaction combined with immunostaining

RNA fluorescence in situ hybridization experiments were performed according to our previously published procedures (Tu et al, 2021). Hybridization chain reaction (HCR) v3.0 method was used to achieve mRNA FISH at high sensitivity and specificity. Probe sets against *Tet* (LOT: PRD615), *dpp* mRNA (LOT: PRD621), *vasa* (LOT: PRG380), and *Act5C* (LOT: PRB668) were ordered from Molecular Instruments, Inc.

## S2 cell transfection and co-immunoprecipitation

Co-IP experiments in S2 cells were performed according to our previously published procedures (Tu et al, 2020). Briefly, S2 cells were grown at 25 °C in the HyClone SFX-Insect Cell Culture Media (SH30278.02, Cytiva). Transfections were performed using the X-treme GENE HP (6366546001, Roche) transfection reagent according to the manufacturer's instructions. Mouse anti-Flag (F1804, Sigma-Aldrich; 1:2000), or mouse anti-HA (H3663, Sigma-Aldrich; 1:2000) antibodies were used for immunoblotting. To avoid interference from the immunoglobulin G (IgG) heavy chain (~55 kDa), horseradish peroxidase–goat anti-mouse IgG light chain secondary antibodies were used. Inputs were extracted before IP.

## Flag-Tet Co-IP and mass spectrometry

For Flag-Tet co-IP, *pAc-Gal4*, and *UASz-Flag-Tet* plasmids were co-transfected into S2 cells. Co-IP was performed with an anti-Flag antibody with an overnight incubation at 4 °C. The beads were first washed four times with 1X IP buffer (50 mM pH 7.5 Tris-HCl, 150 mM NaCl, 0.5% Triton X-100, 1 mM EDTA, and a mixture of protease inhibitors). Then, beads were further washed once with buffer A (1x PBS, 1% NP40, 1 mM EDTA), once with buffer B (buffer A + 0.5 M NaCl), once with buffer C (1xPBS, 1 mM EDTA, 1% NP40), and once with buffer D (1x PBS, 1 mM EDTA). The beads were washed with rotation for 5 min at 4 °C each time. After the washing steps, the bound immune complexes were eluted with 180 μl of 0.1 M glycine (pH 2.8), and immediately neutralized with 20 μl of 1 M Tris-HCl (pH 8.0). Finally, the solution was brought to 400 μl with 100 mM Tris-HCl (pH 8.5), and 100 μl of TCA (100%) was added and well mixed. The samples were kept at 4 °C overnight for precipitating. They were then spun at 14,000 rpm for 30 min at 4 °C, and the supernatant was carefully removed. The pellet was washed twice with 500 μl of cold acetone. The protein pellet was air-dried under a hood. Later, LC/MS acquisition and MS dataset processing were performed as previously described (Mattingly et al, 2022).

## Construction of *Drosophila* stains

For the *Tet-KD2 shRNA* line, we used caagtcgatgattatgcgcaa as the target sequences, and made the *pUAS-mir-Tet-KD2* plasmid according to previous publications (DeLuca and Spradling, 2018; Ni et al, 2011). The plasmid was then inserted at *attp40* site on the second chromosome using PhiC31 integrase-mediated transgenesis. Tet-EGFP was generated using Cas9/CRISPR: a G S linker (ggtggcggcggaagcggaggtggaggctcg), 3XFlag (gactacaaagaccatgacggtgattataaagatcatgacatcgattacaaggatgacgatgacaag), and EGFP cDNA

sequence were inserted at the C-terminus of endogenous *Tet*, just before the stop codon. For making *UASz-Flag-Bap55-Stat92E*, *1XFlag*, *Bap55* CDS (full-length Bap55-RA, FlyBase ID: FBpp0086115), a 6X Alanine link and *Stat92E* CDS (full-length Stat92E-RF, FlyBase ID: FBpp0088489) were cloned into *pUASz1.0*, the plasmid was then inserted at *attP40* site on the second chromosome using PhiC31 integrase-mediated transgenesis. Flag-tagged mutated full-length Tet (RF) cDNA, as indicated in Fig. 2A for respective mutation sites, was cloned into *pUASz1.0* to generate *UASz-Flag-Tet*[WT] and *UASz-Flag-Tet*[ED] constructs. Subsequently, the plasmids were integrated at the attP2 site on the third chromosome utilizing PhiC31 integrase-mediated transgenesis.

## RNA extraction and qPCR

To verify the efficiency of the RNAi strains, we employed the ubiquitous *tub-Gal4* driver combined with *tub-Gal80*[ts] (*tub-Gal4*[ts]) to knock down the target gene for 2 days during the adult stage. Subsequently, we extracted the total RNA from the whole body of flies. Total RNA was extracted using TRIzol™ Reagent (Thermo Fisher Scientific, 15596026), and cDNA was prepared using the PrimeScript™ RT-PCR Kit (Takara, RR014B). qPCR was performed using SYBR Green I Master Mix (Roche, 04707516001). The threshold cycle (CT) values were normalized to Rpl10. Primers used are as follows:

> *Bap55*: tgaccagcgcaagttctatg; tgaatgacattggcgtaggc
> *Bap111*: caagagcaaggtgaagaccg; gcgtaggccaagtgcttggt
> *brm*: atgatgcacccatgctgaag; cttctgttgccgttggttgc
> *Bap170*: tggctggacaacggtctgat; acggactagctggacaatct
> *Bap180*: tgagcatcccattgatctgc; agatctgagaaccaggctcg
> *osa*: tcagcagcaacaagcatcga; atcggataacctcctccagc
> *Rpl10*: atgctaagctgtcgcacaaatg; gttcgatccgtaaccgatgt

## ChIP and qPCR

ChIP was performed following the protocol provided by the Pierce™ Magnetic ChIP Kit (26157, Thermo Fisher Scientific). For each genotype, 300 pairs of ovaries were dissected, and the anterior tips of the germaria were cut with forceps and collected for ChIP experiments. ChIP-grade anti-EGFP (Invitrogen, A11222) and anti-HA (Abcam, ab9110) antibodies were utilized to pull down chromatins. qPCR was conducted using SYBR Green I Master Mix (Roche, 04707516001). The CT values were normalized to IgG negative control. Primers used are as follows:

> *P1*: tctggtctgttcgtactgct; atctgtgtgcgtgtcggtgt
> *P2*: gccgaagtgtgtatctgtgt; cgccgctcatttcggtatag
> *P3*: tggctggcacatctgaacat; tgggatatcgagctgcacga
> *P4*: atgttgctgttgctgctgct; gacactctgtggacgaacga
> *dpp2.0*: ttgggtcagcaacaccagca; agtctgggaaggcactaaag
> *Act5C*: atcgggatggtcttgattctg; actccaaacttccaccactc

## In vitro protein binding assay

The C-terminal domain of Tet (aa1846-2921; Tet-PF, FlyBase ID: FBpp0306013) and the full-length Stat92E (Stat92E-PF, FlyBase ID: FBtr0089486) were cloned into pET32a(+). In addition, Myc and HA tags were inserted at the N-terminus of Tet-C and Stat92E, respectively. After induced expression, bacterial was lysed with

B-PER Bacterial Protein Extraction Reagent (Thermo Fisher Scientific, 90078), and purified with HisPur™ Ni-NTA Resin (Thermo Fisher Scientific, 88221). The full-length Bap55 (Bap55-PA, FlyBase ID: FBpp0086115) was cloned into pFastBac-HTB-2×Strep. Insect baculovirus expressing Strep-Bap55 was generated following the supplier's instructions (Thermo Fisher Scientific, 10584027). Subsequently, the Strep-Bap55 protein was purified using Strep-Tactin XT Superflow Resin (IBA Lifesciences, 24030025) and eluted using Biotin. The eluent containing target protein was applied to a HiLoad 26/600 Superdex 200 pg (Cytiva) that was prepacked column pre-equilibrated with SEC buffer (50 Mm Tris-HCl, pH 7.5, 500 mM NaCl, 1 mM DTT, 1 mM EDTA) for gel-filtration. The fractions containing Strep-Bap55 were collected and concentrated. For the in vitro binding, respective indicated proteins (10 µg of each) were incubated overnight at 4 °C in 200 µl Buffer A (50 mM Tris-HCl, pH 7.5, 50 mM NaCl, 5% Glycerol, 1 mM EDTA, 0.1% Igepal CA-630) with a mixture of protease inhibitors, 40 µl pre-blocked Dynabeads™ Protein G (previously blocked with 0.5% BSA in Buffer A for 2 h at room temperature), and 5 µg antibodies. After six washes with Buffer A, the bound complexes were eluted with 1×SDS sample buffer and subjected to SDS-PAGE and immunoblotting. For pull-down and western blotting, mouse anti-HA (1:2000, Sigma-Aldrich, H3663), mouse anti-Myc (1:2000, Sigma-Aldrich, M5546), or mouse anti-Strep (1:2000, IBA Lifesciences, 2-1507-001) antibodies were employed.

## Quantification and statistical analysis

GSCs and CBs were quantified under the fluorescence microscope according to the method described previously (Ma et al, 2017; Tu et al, 2023). For confocal images, fluorescence intensities for the highlighted areas of interest were quantified using the Leica software or ImageJ, and the mean values of fluorescence intensities and internal controls were collected, as previously described (Shihan et al, 2021; Tu et al, 2020) after subtraction of the background fluorescence. The ratio of mean values of intensities of interest to co-labeled internal controls was calculated, normalized to control samples, and subjected for statistical analysis. All bar graphs are represented as means ± SEM (***$P \leq 0.001$; **$P \leq 0.01$; *$P \leq 0.05$; n.s., no significance). We conducted Student's t-tests using Microsoft Excel or GraphPad Prism 9 for statistical analysis. The analysis was specifically performed to compare between two groups, as indicated by the arrows in the graphs. The source data for the analysis are available online.

## Data availability

This study includes no data deposited in external repositories.

## Peer review information

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

## Acknowledgements

We would like to thank Xie laboratory members for advice and discussion, Bloomington *Drosophila* Stock Center, the TRiP at Harvard Medical School (NIH/NIGMS R01-GM084947), *Drosophila* Genomics Resource Center (NIH Grant 2P40OD010949), Developmental Studies Hybridoma Bank, HKUST BioCRF Facility, Dr. Margret Shirinian and Dr. Ruth Steward for reagents. This work was supported by Stowers Institute for Medical Research (TX), a grant from National Institutes of Health (R01HD097664 to TX), and the grants from Hong Kong Research Grants Council (TRS_T13-602/21-N and GRF_16104621 to TX, GRF_16103822 to RT), and grants from the National Natural Science Foundation of China (No. 31870746), Shenzhen Basic Research Grants (JCYJ20200109140414636) and Natural Science Foundation of Guangdong Province, China (No. 2021A1515010796 and 2022A1515010666) to WL.

## Author contributions

**Renjun Tu**: Conceptualization; Data curation; Formal analysis; Funding acquisition; Validation; Investigation; Visualization; Methodology; Writing—original draft; Writing—review and editing. **Zhaohua Ping**: Conceptualization; Data curation;

Formal analysis; Investigation; Visualization. **Jian Liu**: Investigation; Methodology. **Man Lung Tsoi**: Investigation. **Xiaoqing Song**: Investigation. **Wei Liu**: Resources; Supervision; Funding acquisition; Investigation; Methodology. **Ting Xie**: Conceptualization; Resources; Data curation; Formal analysis; Supervision; Funding acquisition; Investigation; Methodology; Writing—original draft; Project administration; Writing—review and editing.

## Disclosure and competing interests statement

The authors declare no competing interests.

# Expanded View Figures

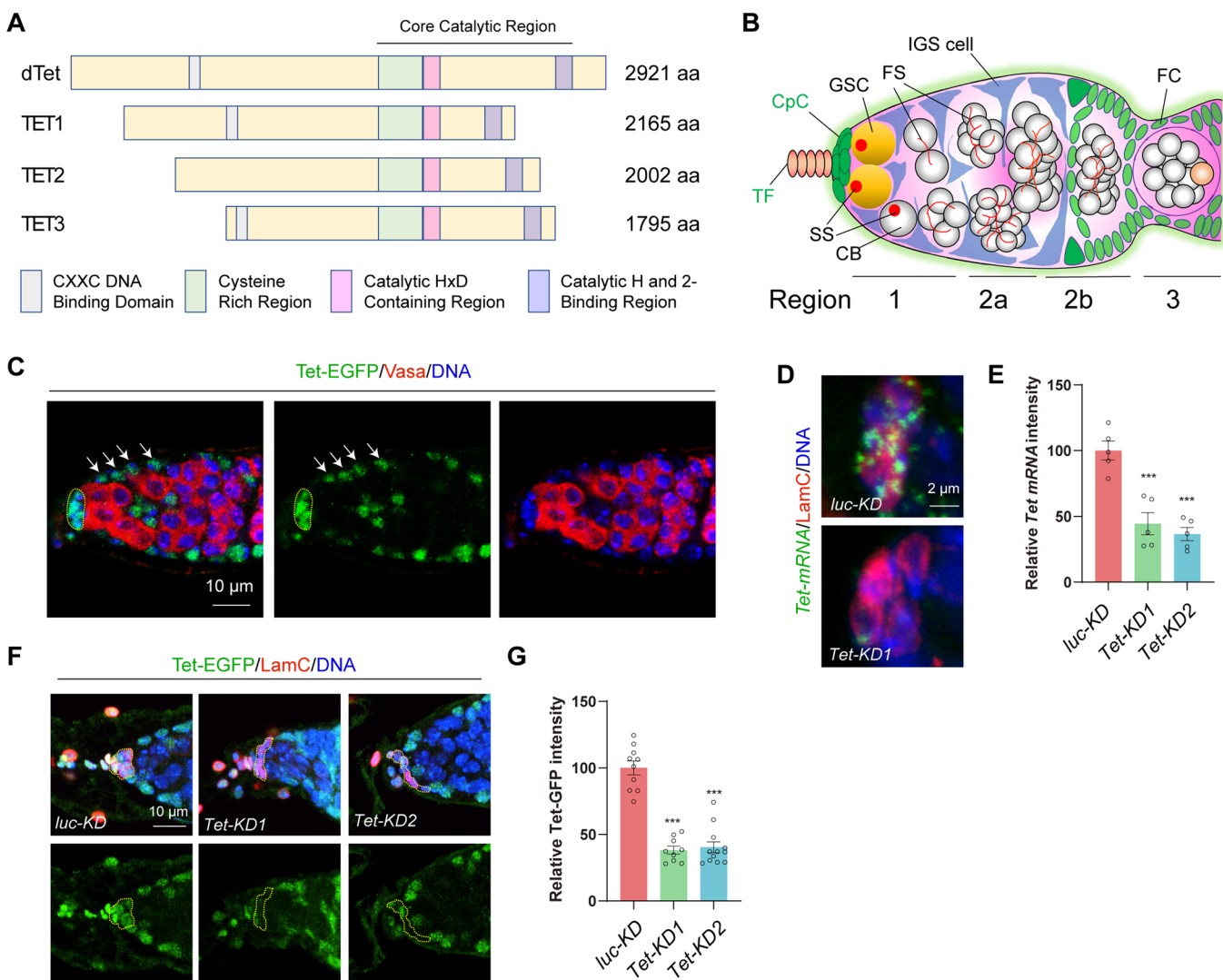

**Figure EV1.   Validating the knockdown efficiency of Tet RNAi lines.**

(A) Domain architecture of *Drosophila* Tet (NP_001261344.1), human TET1 (NP_001393294.1), TET2 (NP_001120680.1), and TET3 (NP_001274420.1). They share the highly conserved CXXC DNA binding domain, the cysteine-rich region, and the catalytic domain regions. (B) A schematic diagram of a *Drosophila* germarium, which contains GSCs, cystoblasts (CBs), mitotic cysts (2-cell, 4-cell, and 8-cell cysts) and 16-cell cysts, and stage 1 egg chamber in region 1, 2a, 2b, and 3. Abbreviations: TF, terminal filament; CPC, cap cell; IGS cell, inner germarial sheath cell; CB, cystoblast; SS, spectrosome; FS, fusome; and FC, follicle cells. (C) Confocal images show that Tet-EGFP (arrows) does not express in Vasa-labeled (Red) germ cells. (D–G) *bab1*^ts^-driven *Tet-KD* significantly reduces *Tet mRNA* (D) and Tet-EGFP (F) expression in niche cells. (E, G) Quantification results. Data information: In (E and G), data are presented as mean ± SEM. ***$P \leq 0.001$ (Student's *t*-test). $n$ = number of germaria. Source data are available online for this figure.

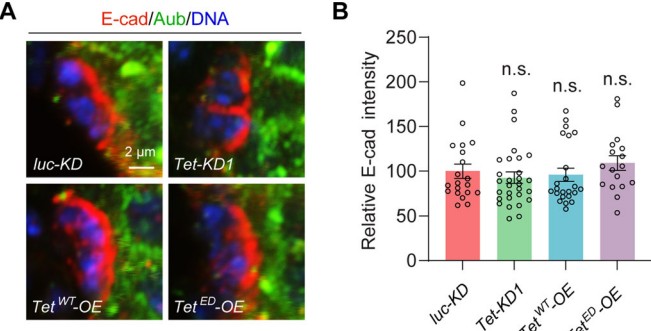

**Figure EV2. Tet is dispensable in the niche for maintaining E-cadherin accumulation at the GSC-niche junction.**

(A, B) *bab1^{ts}*-driven *Tet-KD* does not have an obvious effect on E-cadherin (E-cad) expression. The Argonaute/Piwi family protein, Aubergine (Aub), is enriched in germ cells. anti-Aub antibody was used to label germ cells. (B) Quantification results (*n* = number of germaria). Data are presented as mean ± SEM. n.s., no significance (Student's t-test). Source data are available online for this figure.

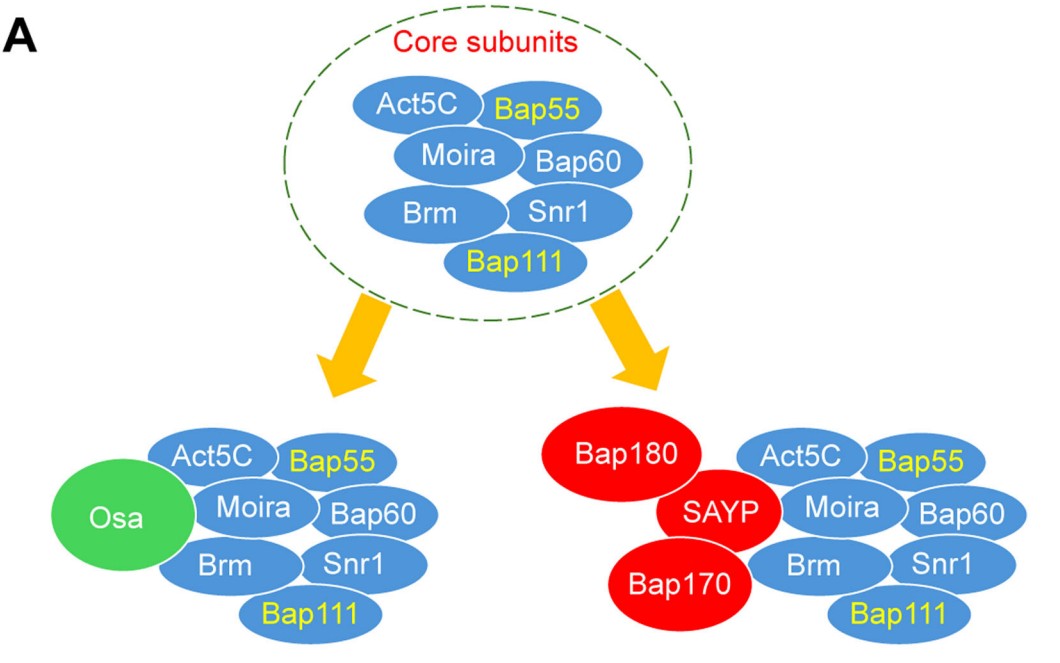

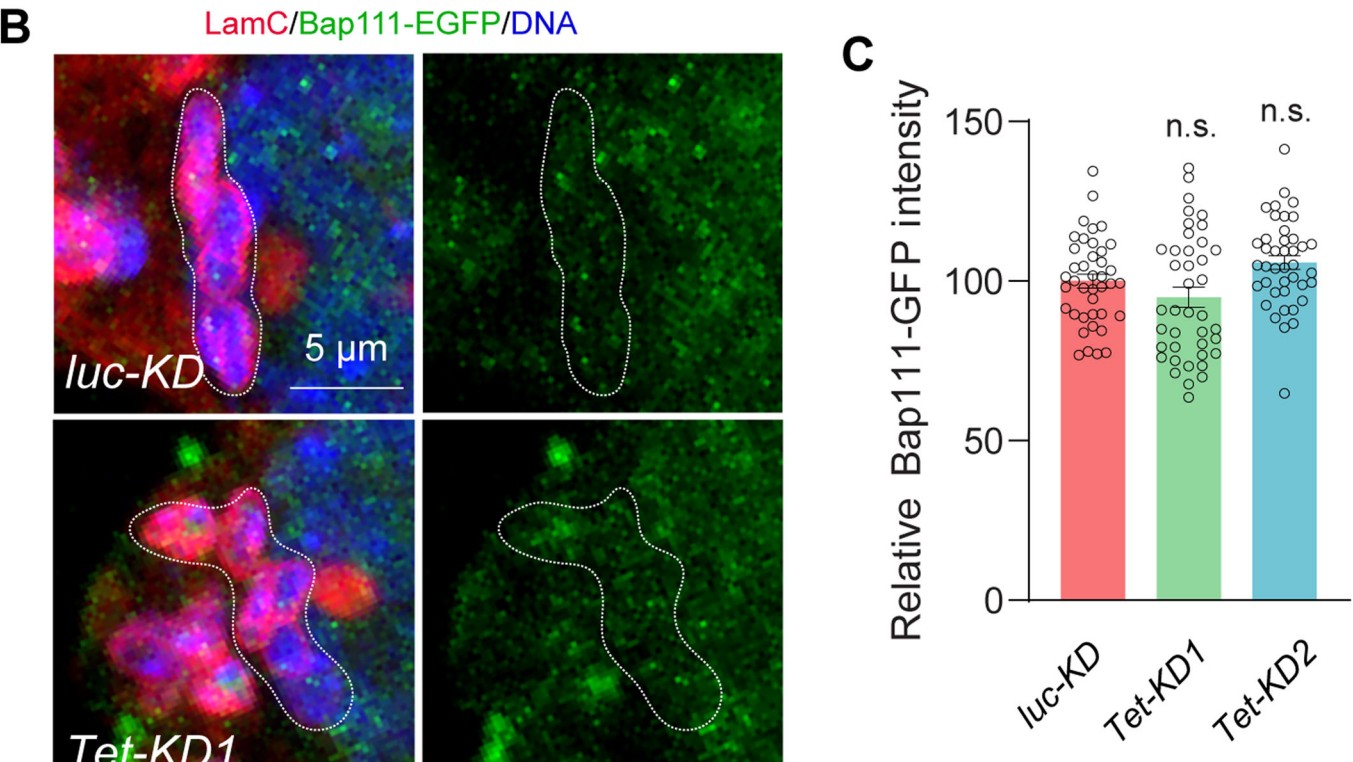

**Figure EV3.  Tet is dispensable in the niche for maintaining Bap111-EGFP expression.**

(A) The schematic drawing of Brm complexes. Brm-containing chromatin remodelers can be divided into two types: BAP and PBAP according to their specific subunits, Osa and Bap170/180/SAYP, respectively (Hong and Choi, 2016). (B, C) *bab1^{ts}*-driven *Tet-KD* does not have an obvious effect on Bap111-EGFP expression in niche cells. (C) Quantification results (*n* = number of germaria). Data are presented as mean ± SEM. n.s., no significance (Student's t-test). Source data are available online for this figure.

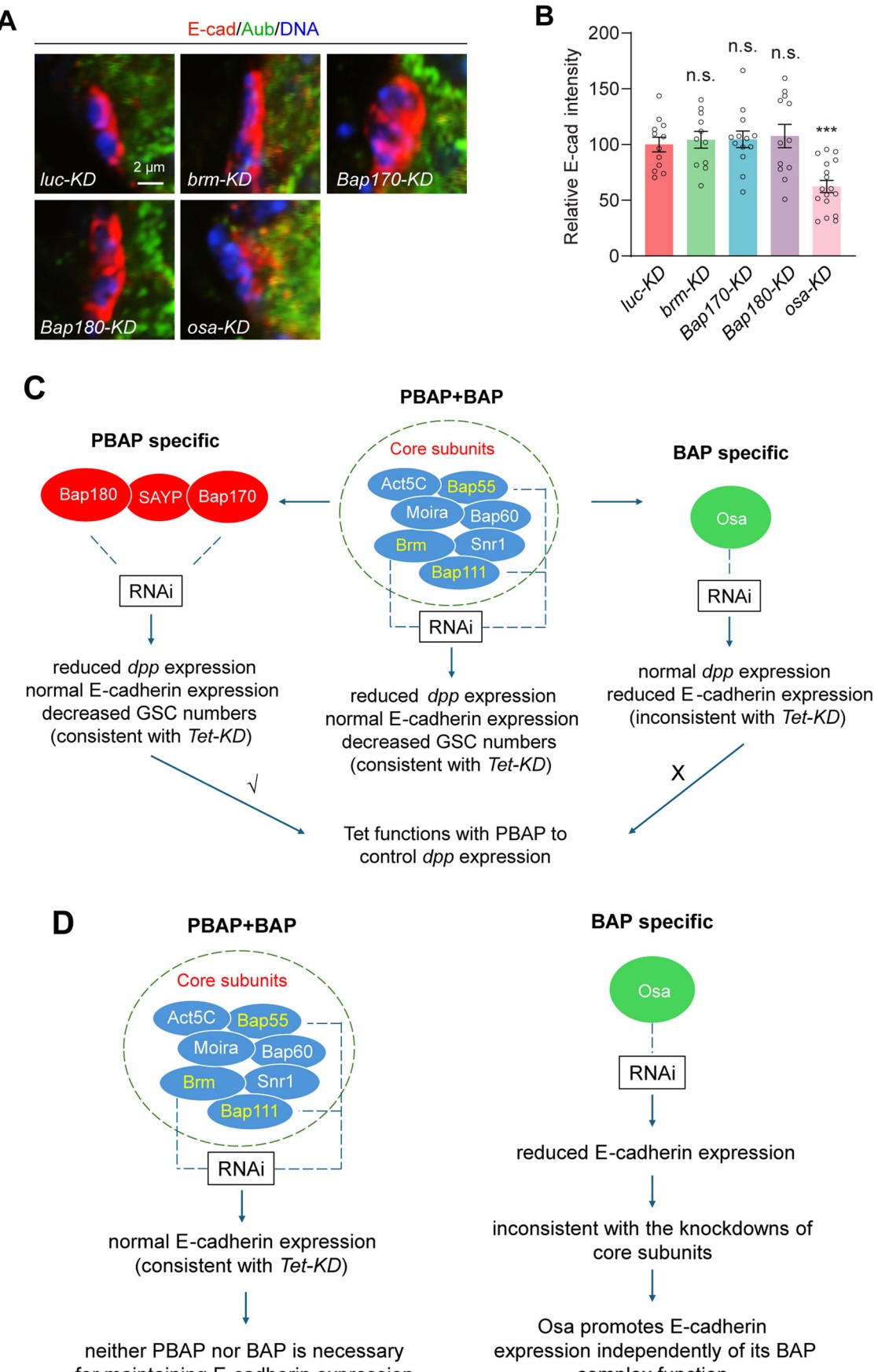

◀  **Figure EV4.  Osa is required in the niche for E-cadherin expression.**

(A, B) *bab1^ts*-driven *brm-KD*, *Bap170-KD*, and *Bap180-KD* have no obvious effect on E-cadherin expression in cap cells, but *osa-KD* significantly decreases E-cad expression in niche cells. (B) Quantification results (*n* = number of germaria). Data are presented as mean ± SEM. n.s., no significance (Student's t-test). (C) Knockdowns of PBAP-specific but not BAP-specific subunits exhibit similar phenotypes compared to *Tet-KD* and core subunits-*KD*. (D) Knockdowns of core subunits have no effect on the expression of E-cadherin, indicating that neither PBAP nor BAP is required for maintaining E-cadherin expression. The reduced expression level of E-cadherin observed in *osa-KD* germaria is likely caused by the function of Osa independent of BAP complex. Further experiments need to be conducted in the future to verify this intriguing phenomenon. Source data are available online for this figure.

