## [Peer Review File · The EMBO Journal]

Niche Tet maintains germline stem cells independently of dioxygenase activity

Renjun Tu, Zhaohua Ping, Jian Liu, Man Lung Tsoi, Xiaoqing Song, Wei Liu, and Ting Xie

Corresponding author: Ting Xie (tgx@ust.hk)

Review Timeline:

Submission Date:	12th Sep 23
Editorial Decision:	30th Oct 23
Revision Received:	28th Dec 23
Editorial Decision:	6th Feb 24
Revision Received:	19th Feb 24
Accepted:	23rd Feb 24

Editor: Daniel Klimmeck

Transaction Report:

Dear Dr Xie,

Thank you again for submitting your manuscript EMBOJ-2023-115586 for consideration by the EMBO Journal. Please accept my sincere apologies for getting back to you with this unusual protraction due to delayed referee input, as well as detailed discussion in the editorial team. As indicated, your manuscript has been seen by three referees with expertise in germline biology and stem cell chromatin biology, and we have received reports from all of them, which are shown below.

As you will see from their comments, the referees acknowledge the potential interest and value of your findings for the field. However, they also express major concerns, which need to be addressed thoroughly to make them supportive of publication in the EMBO Journal. In more detail, the referee #3 points to some precedence on Tet dioxygenase independent function and states that the interplay between Tet, The BPAP complex and STAT92 as well as targeting of dpp is too prematurely addressed (ref#2, standfirst; also ref#1 pt. 1). In addition, reviewer #3 points to data inconsistencies which need to be resolved (ref#3. pt.1). Further, the referees raise a number of points related to methods annotation, additional controls required, data presentation, statistics applied and discussion of the results in the manuscript as well as literature references that would need to be conclusively addressed to achieve the level of robustness and clarity needed for The EMBO Journal.

Given the overall interest stated and broader angle of your findings, we are able to invite you to revise your manuscript experimentally to address the referees' comments. I need to stress though that we do require strong support from the referees on a revised version of the study in order to move on to publication of the work.

In light of the extensive experimentation requested, I would appreciate if you could contact me during the next weeks for exchange e.g. a video call to discuss your perspective on the comments and potential plan for revisions.

Please feel free to contact me if you have any questions or need further input on the referee comments.

When submitting your revised manuscript, please carefully review the instructions below.

Please feel free to approach me any time should you have additional questions related to this.

Thank you for the opportunity to consider your work for publication.

I look forward to your revision.

Kind regards,

Daniel Klimmeck

Daniel Klimmeck, PhD
Senior Editor
The EMBO Journal

Instruction for the preparation of your revised manuscript:

- 1) a .docx formatted version of the manuscript text (including legends for main figures, EV figures and tables). Please make sure that the changes are highlighted to be clearly visible.
- 2) individual production quality figure files as .eps, .tif, .jpg (one file per figure).
- 3) a .docx formatted letter INCLUDING the reviewers' reports and your detailed point-by-point response to their comments. As part of the EMBO Press transparent editorial process, the point-by-point response is part of the Review Process File (RPF), which will be published alongside your paper.

4) a complete author checklist, which you can download from our author guidelines ([https://wol-prod-cdn.literatumonline.com/pb-assets/embo-site/Author Checklist%20-%20EMBO%20J-1561436015657.xlsx](https://wol-prod-cdn.literatumonline.com/pb-assets/embo-site/Author%20Checklist%20-%20EMBO%20J-1561436015657.xlsx)). Please insert information in the checklist that is also reflected in the manuscript. The completed author checklist will also be part of the RPF.

6) It is mandatory to include a 'Data Availability' section after the Materials and Methods. Before submitting your revision, primary datasets produced in this study need to be deposited in an appropriate public database, and the accession numbers and database listed under 'Data Availability'. Please remember to provide a reviewer password if the datasets are not yet public (see <https://www.embopress.org/page/journal/14602075/authorguide#datadeposition>).

7) Our journal encourages inclusion of *data citations in the reference list* to directly cite datasets that were re-used and obtained from public databases. Data citations in the article text are distinct from normal bibliographical citations and should directly link to the database records from which the data can be accessed. In the main text, data citations are formatted as follows: "Data ref: Smith et al, 2001" or "Data ref: NCBI Sequence Read Archive PRJNA342805, 2017". In the Reference list, data citations must be labelled with "[DATASET]". A data reference must provide the database name, accession number/identifiers and a resolvable link to the landing page from which the data can be accessed at the end of the reference. Further instructions are available at .

8) At EMBO Press we ask authors to provide source data for the main and EV figures. Our source data coordinator will contact you to discuss which figure panels we would need source data for and will also provide you with helpful tips on how to upload and organize the files.

Numerical data can be provided as individual .xls or .csv files (including a tab describing the data). For 'blots' or microscopy, uncropped images should be submitted (using a zip archive or a single pdf per main figure if multiple images need to be supplied for one panel). Additional information on source data and instruction on how to label the files are available at .

9) We replaced Supplementary Information with Expanded View (EV) Figures and Tables that are collapsible/expandable online (see examples in <https://www.embopress.org/doi/10.15252/emboj.201695874>). A maximum of 5 EV Figures can be typeset. EV Figures should be cited as 'Figure EV1, Figure EV2' etc. in the text and their respective legends should be included in the main text after the legends of regular figures.

11) For data quantification: please specify the name of the statistical test used to generate error bars and P values, the number (n) of independent experiments (specify technical or biological replicates) underlying each data point and the test used to calculate p-values in each figure legend. The figure legends should contain a basic description of n, P and the test applied. Graphs must include a description of the bars and the error bars (s.d., s.e.m.).

Please remember: Digital image enhancement is acceptable practice, as long as it accurately represents the original data and conforms to community standards. If a figure has been subjected to significant electronic manipulation, this must be noted in the

figure legend or in the 'Materials and Methods' section. The editors reserve the right to request original versions of figures and the original images that were used to assemble the figure.

We realize that it is difficult to revise to a specific deadline. In the interest of protecting the conceptual advance provided by the work, we recommend a revision within 3 months (28th Jan 2024). Please discuss the revision progress ahead of this time with the editor if you require more time to complete the revisions.

Referee #1:

In this study, the authors investigate the role of the Tet protein in *Drosophila* ovarian germline stem cell maintenance. They find that Tet is expressed in somatic cells and that RNAi knockdown specifically in the *bab1+* population (which provide niche function for the GSCs) impairs GSC self-renewal. Further investigation reveals that this is due to a non-enzymatic scaffolding function of the tet protein that brings *stat92E* together with the BAP/PBAP complex protein Bap55. Building on a previous study that found that Jak/Stat signaling activates *dpp* expression in the *bab1+* population, they show that RNAi knockdown of *tet* or the PBAP proteins (but not *osa*, which is specific for the BAP complex) reduces *dpp* expression, BMP signaling in GSCs, and GSC number. Rescue experiments using a wildtype or enzymatic dead allele of *tet* and in vitro co-IP experiments provide substantial support for the idea that *tet* has a non-enzymatic, scaffolding role in this process. Taken together, their data advance our understanding of the molecular mechanisms that contribute to *Drosophila* germline stem cell self renewal and the evidence for a non-enzymatic function of *tet* will be of interest more broadly. However, several concerns should be addressed before publication:

1. The data provided suggest that *osa* is required for GSC maintenance but not for *dpp* expression of BMP signaling within GSCs. What is the explanation for this? Also, since the conclusion from these data that the BAP complex is not required for *dpp* expression is based on this one RNAi line, the authors should confirm that the *osa* RNAi is, indeed, effectively knocking down *osa* transcript, as expected.
2. The mass spectrometry approach associated with the data in Fig. 3A do not appear to be described in the methods. This should be added.
3. Throughout the figures, it is not clear what is meant by "relative intensity". What are these intensity measurements relative to? The methods state that internal imaging controls were measured, but do not provide details about what these internal controls are. Also, the method of controlling for background signal (e.g. a background subtraction or correction) should be described.
4. The method for unambiguously identifying CBs is not well described. Related to this, the object that the arrowhead in Fig. 1C is pointing to is not a very clear example of a rounded spectrosome that is clearly distinct from the other hts structures nearby.
5. In Fig. S3, the Bap111 channel should be shown separately. The signal is dim and hard to appreciate in the multicolor image. In addition, these data show that *tet* is not essential for maintaining *bap111* expression but do not provide strong support for the statement that Tet is not essential for maintaining the expression level of the whole BAP and PBAP complexes, as stated at the bottom of page 8.

Referee #2:

In this work, the authors report the function of the Tet protein in the *Drosophila* ovarian stem cell niche. They convincingly show that an enzymatic-dead Tet is necessary to ensure proper *dpp* transcription, an essential gene for germline stem cell (GSC) maintenance in the niche. Tet is required to bring together components of the PBAP complex and the Stat92E transcription factor. In their model, complexed Tet-PBAP-Stat bind to the *dpp* gene to activate its transcription and, thus, to regulate niche activity.

I believe the biological question being investigated to be of interest. However, in my opinion the novelty of the findings appears somewhat limited as to warrant publication in a journal with a broad readership such as The EMBO J. Dioxygenase activity-independent Tet functions are well established in the field, as is the relationship between Jak/Stat and *dpp* transcription in the GSC niche. The MS does report some advancements, such as the demonstration of Tet-Stat92 and Tet-Bap55 physical interactions and the role of the complex in regulating *dpp* mRNA levels in the cap cells. Nevertheless, it fails to provide evidence for the binding of the complex to the *dpp* gene.

Major points:

- 1- I have a serious issue with the current presentation of the MS. To follow is a list (by no means exhaustive) of some of the problems I detected:
 - a) I have found typos and spelling mistakes in the text and figures (for instance, Fig 1L "reseuce" instead of "rescue"; Fig 2A enzyme, not ensyme).
 - b) In several instances, the authors talk of "decrease GSCs" or "comparable CBs". I assume they refer to reduced GSC

numbers or comparable CB numbers.

c) Sentences with very similar contents are repeated throughout the text (for instance, in the Abstract: "Tet acts as a scaffold protein, bringing together the PBAP chromatin remodeling complex and the Stat92E transcription factor to activate the expression of BMP-like Dpp in the niche" and, two sentences later, "Mechanistically, Tet interacts directly with BAP55 and Stat92E, facilitating recruitment of the PBAP complex to the dpp promoter and activating Dpp expression". Or in the first paragraph of the Discussion "Tet functions as a protein scaffold to recruit the PBAP complex to Stat92E by directly interacting with Bap55 and Stat92E. Bap55-Stat92E can bypass the requirement of Tet in the niche to maintain BMP signaling and GSC self-renewal" and later in the paragraph "we propose that Tet acts as a protein scaffold, functioning independently of its enzymatic activity. It facilitates the interaction between the PBAP complex and Stat92E, leading to the activation of dpp expression in the niche". Or later in the Discussion "In this study, we show that TET functions as a protein scaffold independently of its enzymatic activity to recruit the PBAP complex to Stat92E to activate Dpp expression in the niche for controlling GSC self-renewal in the Drosophila ovary. In this study, we have used a combination of genetics and biochemistry to demonstrate that Tet functions independently of its dioxygenase activity in the niche to control GSC self-renewal by regulating BMP signaling".

d) There is one duplicated reference (Delatte et al 2016).

e) Fig. S1 is missing from this version.

f) The writing of the Fig. legends lacks detail and is redundant (every single panel containing statistical analyses has the following (or very similar) statement "Student's t test: *** $p < 0.001$; n. s., no significance". Or the repetition of the meaning of a dash line, arrows or arrowheads in all the panels of the same figure. This makes fig. legends very repetitive)

g) Gene nomenclature is not consistent throughout (for instance, there is Bab1-Gal4, bab-Gal4, bab1-Gal4; or there is tet-RR-wt and tet-wt, tet-Dead and tet-ED)

h) The full genotypes of the different experiments are not reported (what is "wt" in the different experiments? Canton S or Oregon flies? or Gal4 lines without UAS transgenes, or viceversa?).

i) Treatments should also be clearly reported. The M+M section has a general statement that does not provide sufficient detail "newly eclosed flies at room temperature were cultured at 29{degree sign}C for the specified days before phenotypic analysis". As you may hint from my words, I am disappointed and frustrated to read a text in such need of a profound grammar and style revision. In my opinion, the current version of the MS makes a disservice to its scientific contents.

2- In the Abstract, the authors mention that "Tet interacts directly with BAP55 and Stat92E, facilitating recruitment of the PBAP complex to the dpp promoter and activating Dpp expression". However, they do not demonstrate any binding of the complex to the dpp gene. I believe the work would be much improved if the authors could show the binding of the complex to the dpp promoter or to an enhancer of the gene. I would suggest to do chromatin immunoprecipitation with S2 cells or similar, provided they express dpp and they respond to tet-stat92E-bap55 interactions (they use this cell type to search for tet-associated proteins, Fig. 3A). Alternatively, they could try an EMSA (electrophoretic mobility shift assay) with suitable dpp sequences.

Minor points:

1- The Title of the MS reads "A novel dioxygenase-independent tet function...". I am not sure if this is really a novel function. I have found several published articles in which some of the vertebrate Tet genes are required to complex transcription factors to their target genes.

2- Please explain the meaning of IGS. Do you refer to escort cells?

3- "In the germarium, two or three GSCs directly contact cap cells and anterior IGS cells, which form the niche for controlling their self-renewal, whereas CBs, mitotic cysts and early 16-cell cysts are wrapped up by the long cellular processes of posterior IGS cells, which constitute the niche for promoting these GSC progeny differentiation". Consider improving the grammar. Also, which renewal are you refereeing to? I assume it is GSC renewal, but it is not clear from the sentence.

4- "Furthermore, Jak-Stat signaling controls Dpp expression in cap cells to maintain GSC self renewal". Please cite, here and in the rest of the text, Lopez-Onieva et al. Development 2008 (Jak/Stat signalling in niche support cells regulates dpp transcription to control germline stem cell maintenance in the Drosophila ovary).

5- "In this study, two independent transgenic shRNA lines were used to knock down Tet in adult cap cells (bab1ts>Tet-KD1 and bab1ts>Tet-KD2)". Careful with this statement (and also in other sections of the text). In addition to cap cells, bab1-Gal4 is also expressed in terminal filament cells and in escort cells.

6- I would not claim that LamC is "specifically expressed in TF and cap cells". It is enriched in these cell types, while also presenting lower levels in other cells of the germarium.

7- Arrowhead in Fig. 1K seems to be pointing to the ring-canal associated fusome material of a 4-cell cyst recently divided, not to a CB spectrosome.

8- When referring to the BAC transgenic strain for Bap111-GFP, please specify. Consider stating that Bap111-GFP is an in-frame insertion in the genomic locus of a BAC transgene. Therefore, the flies have the two endogenous bap111 copies plus the BAC's babp111-GFP.

9- "To determine which complex is involved in controlling dpp expression, we sought to knock down the expression of brm, Bap170, Bap180, and osa in adult niche cells, brm-KD, Bap170-KD, Bap180KD, and osa-KD. brm-KD, Bap170-KD, Bap180KD, and osa-KD significantly decrease GSCs compared to the control". Please amend.

10- Fig. 5B: Why does the input Flag-tet-CF-1 have so little amount compared to the other lines? Please explain in the fig.

legend, particularly so when the IPed Flag-tet-CF-1 line has plentiful. Why does the IPed Flag-tet-CF-4 line show two bands?

11- Fig. 5D, F: why are there so many bands of Myc-Tet-C (D, E) and HA-Stat92E (F)? What do the asterisks mean in D-F?

12- Bap55-Stat92E fusion: How was this fusion made? Details in the M+M are scarce and confusing, at least to me, as I do not know if it is just an in-frame fusion of both full length ORFs or something different. This is one of the most important experiments of the MS and should be reported with greater detail.

13- Fig. 5G: I assume the difference in the number of GSCs/germarium of wt and tet-KD-1 is statistically significant. The authors

may want to show that in the graph, as they have done with 5K and 5L.

14- Human tet3: It would be interesting to show if a dead version of the Tet3 protein is also able to rescue.

Referee #3:

This manuscript demonstrates a novel function of TET proteins, independent of catalytic activity, to act as a scaffold protein the associate the PBAP chromatin remodelling complex with STAT92E in order to activate expression of Dpp in the Drosophila ovarian stem cell niche. This is a highly significant study from 2 aspects: 1) association of a novel TET activity (and also showing conservation of function between Drosophila and human) but also 2) it has been long known that STAT and Dpp signalling somehow interact in the stem cell niche but the mechanism has been unclear. This study provides mechanistic evidence for this interaction. The manuscript is well written and data are clearly presented.

Concerns that need to be addressed:

- 1) The authors state "The Tet-KD1 and Tet-KD2 germaria significantly decrease GSCs compared to the control although the control and Tet knockdown germaria have comparable CBs (Fig. 1C and 1D)." Why does a loss of GSCs also not flow on the result in a loss of CBs?
- 2) The Materials and Methods state that all of the statistical tests are T-tests. Given that the graphs all show more than two groups should the statistical tests not all have been ANOVAs?
- 3) The authors state "Like Tet overexpression, niche-specific hTET3 overexpression (hTET3-OE) alone can slightly increase dpp mRNA expression in cap cells and pMad expression in GSCs but does not have obvious effect on GSCs (Fig. 6A-F)." However, Fig. 6F does not show an increase in the hTet3-OE column compared to WT control, and 6D does not show an increase in pMad expression compared to WT.
- 4) In the discussion could the authors please provide further statements regarding the interaction between the PBAP complex and STAT92E. Does STAT92E need to be prior phosphorylated via JAK activity before interacting with STAT92E.

Additional minor concerns are:

- 1) In Figure 1B could a germ cell marker also be shown to conclusively show that TET-EGFP is not expressed in germ cells

Referee #1, additional comment:

One point for discussion is the reviewer comment about T-tests vs ANOVA. Although the figures do show quantification of more than two groups, the relevant statistical tests are the specific pairwise comparisons between wildtype vs each of the experimental genotypes. ANOVA would be more appropriate if the experimenters had no a priori knowledge of which pairwise comparison would be most relevant or interesting (e.g. a difference between two of the experimental genotypes would be just as important as a difference between a control and an experimental). Since this is not the case, I believe T-tests are most appropriate. However, since they are performing multiple T-tests in the same experiment, one could argue that a multiple comparison correction such as bonferroni should be applied, but I don't think that is commonly done in this case.

Referee #1:

In this study, the authors investigate the role of the Tet protein in Drosophila ovarian germline stem cell maintenance. They find that Tet is expressed in somatic cells and that RNAi knockdown specifically in the bab1+ population (which provide niche function for the GSCs) impairs GSC self-renewal. Further investigation reveals that this is due to a non-enzymatic scaffolding function of the tet protein that brings stat92E together with the BAP/PBAP complex protein Bap55. Building on a previous study that found that Jak/Stat signaling activates dpp expression in the bab1+ population, they show that RNAi knockdown of tet or the PBAP proteins (but not osa, which is specific for the BAP complex) reduces dpp expression, BMP signaling in GSCs, and GSC number. Rescue experiments using a wildtype or enzymatic dead allele of tet and in vitro co-IP experiments provide substantial support for the idea that tet has a non-enzymatic, scaffolding role in this process. Taken together, their data advance our understanding of the molecular mechanisms that contribute to Drosophila germline stem cell self renewal and the evidence for a non-enzymatic function of tet will be of interest more broadly. However, several concerns should be addressed before publication:

Response: Thank this reviewer for recognizing the significance and importance of our study and providing some constructive comments.

1. The data provided suggest that osa is required for GSC maintenance but not for dpp expression of BMP signaling within GSCs. What is the explanation for this? Also, since the conclusion from these data that the BAP complex is not required for dpp expression is based on this one RNAi line, the authors should confirm that the osa RNAi is, indeed, effectively knocking down osa transcript, as expected.

Response: There are two BRM-containing chromatin remodeling complexes, PBAP and BAP, in *Drosophila*. Although PBAP and BAP components are required to maintain GSCs, knocking down the PBAP/BAP-shared components and PBAP-specific component disrupts BMP signaling in GSCs but not E-cadherin accumulation in the stem cell-niche junction. However, knocking down Osa causes the downregulation of E-cadherin in the stem cell-niche junction, suggesting that Osa regulates E-cadherin expression in the niche independently of the BAP complexes. Using qPCR, we have confirmed that the *osa* RNAi strain effectively reduces *osa* mRNA levels when driven by *tub-Gal4* (Fig. S1F).

2. The mass spectrometry approach associated with the data in Fig. 3A do not appear to be described in the methods. This should be added.

Response: The information has been added to the revised manuscript as requested.

3. Throughout the figures, it is not clear what is meant by "relative intensity". What are these intensity measurements relative to? The methods state that internal imaging controls were measured, but do not provide details about what these internal controls are. Also, the method of controlling for background signal (e.g. a background subtraction or correction) should be described.

Response: The term "relative intensity" in this context is referred to as the intensity to the control. We have included a description of these procedures in the Methods section.

4. The method for unambiguously identifying CBs is not well described. Related to this, the object that the arrowhead in Fig. 1C is pointing to is not a very clear example of a rounded spectrosome that is clearly distinct from the other hts structures nearby.

Response: CBs are defined as single germ cells with a round fusome and at least one cell distance from cap cells. We have replaced Fig. 1C with high quality images.

5. In Fig. S3, the Bap111 channel should be shown separately. The signal is dim and hard to appreciate in the multicolor image. In addition, these data show that tet is not essential for maintaining bap111 expression but do not provide strong support for the statement that Tet is not essential for maintaining the expression level of the whole BAP and PBAP complexes, as stated at the bottom of page 8.

Response: We have included single-channel images as suggested. We agree that Bap111-GFP does not fully represent overall expression levels of the entire PBAP complex. Thus, we have revised our statement accordingly.

Referee #2:

In this work, the authors report the function of the Tet protein in the Drosophila ovarian stem cell niche. They convincingly show that an enzymatic-dead Tet is necessary to ensure proper dpp transcription, an essential gene for germline stem cell (GSC) maintenance in the niche. Tet is required to bring together components of the PBAP complex and the Stat92E transcription factor. In their model, complexed Tet-PBAP-Stat bind to the dpp gene to activate its transcription and, thus, to regulate niche activity.

I believe the biological question being investigated to be of interest. However, in my opinion the novelty of the findings appears somewhat limited as to warrant publication in a journal with a broad readership such as The EMBO J. Dioxygenase activity-independent Tet functions are well established in the field, as is the relationship between Jak/Stat and dpp transcription in the GSC niche. The MS does report some advancements, such as the demonstration of Tet-Stat92 and Tet-Bap55 physical interactions and the role of the complex in regulating dpp mRNA levels in the cap cells. Nevertheless, it fails to provide evidence for the binding of the complex to the dpp gene.

Response: We appreciate the recognition of the significance of our work and the constructive comments from this reviewer.

Major points:

1- I have a serious issue with the current presentation of the MS. To follow is a list (by no means exhaustive) of some of the problems I detected:

a) I have found typos and spelling mistakes in the text and figures (for instance, Fig 1L "reseuce" instead of "rescue"; Fig 2A enzyme, not ensyme).

b) In several instances, the authors talk of "decrease GSCs" or "comparable CBs". I assume they refer to reduced GSC numbers or comparable CB numbers.

c) Sentences with very similar contents are repeated throughout the text (for instance, in the Abstract: "Tet acts as a scaffold protein, bringing together the PBAP chromatin remodeling complex and the Stat92E transcription factor to activate the expression of BMP-like Dpp in the niche" and, two sentences later, "Mechanistically, Tet interacts directly with BAP55 and Stat92E, facilitating recruitment of the PBAP complex to the dpp promoter and activating Dpp expression". Or in the first paragraph of the Discussion "Tet functions as a protein scaffold to recruit the PBAP complex to Stat92E by directly interacting with Bap55 and Stat92E. Bap55-Stat92E can bypass the requirement of Tet in the niche to maintain BMP signaling and GSC self-renewal" and later in the paragraph "we propose that Tet acts as a protein scaffold, functioning independently of its enzymatic activity. It facilitates the interaction between the PBAP complex and Stat92E, leading to the activation of dpp

expression in the niche". Or later in the Discussion "In this study, we show that Tet functions as a protein scaffold independently of its enzymatic activity to recruit the PBAP complex to Stat92E to activate Dpp expression in the niche for controlling GSC self-renewal in the Drosophila ovary. In this study, we have used a combination of genetics and biochemistry to demonstrate that Tet functions independently of its dioxygenase activity in the niche to control GSC self-renewal by regulating BMP signaling".

d) There is one duplicated reference (Delatte et al 2016).

e) Fig. S1 is missing from this version.

*f) The writing of the Fig. legends lacks detail and is redundant (every single panel containing statistical analyses has the following (or very similar) statement "Student's t test: *** $p < 0.001$; n. s., no significance". Or the repetition of the meaning of a dash line, arrows or arrowheads in all the panels of the same figure. This makes fig. legends very repetitive)*

g) Gene nomenclature is not consistent throughout (for instance, there is Bab1-Gal4, bab-Gal4, bab1-Gal4; or there is tet-RR-wt and tet-wt, tet-Dead and tet-ED)

h) The full genotypes of the different experiments are not reported (what is "wt" in the different experiments? Canton S or Oregon flies? or Gal4 lines without UAS transgenes, or viceversa?).

i) Treatments should also be clearly reported. The M+M section has a general statement that does not provide sufficient detail "newly eclosed flies at room temperature were cultured at 29{degree sign}C for the specified days before phenotypic analysis".

As you may hint from my words, I am disappointed and frustrated to read a text in such need of a profound grammar and style revision. In my opinion, the current version of the MS makes a disservice to its scientific contents.

Response: Thank the reviewer for taking time to identify errors and provide suggestions. We have made the following revisions as suggested: correcting the typos (1a, 1b), reversing the statements for clarity (1c), adding the reference (1d), resolving the genotype notation issue (1g). Regarding point 1e, the absence of Fig S1 in the reviewer's download might be due to an issue in the journal's website. Regarding point 1f, we acknowledge that our initial submission did not closely follow the EMBO format. In the new version, we have updated the figure legends accordingly. For point 1h, we used *luc-KD* as the control, and we have provided a description of this in the revised manuscript. Concerning point 1i, we have added a detailed description as suggested.

2- In the Abstract, the authors mention that "Tet interacts directly with BAP55 and Stat92E, facilitating recruitment of the PBAP complex to the dpp promoter and activating Dpp expression". However, they do not demonstrate any binding of the complex to the dpp gene. I believe the work would be much improved if the authors could show the binding of the complex to the dpp promoter or to an enhancer of the gene. I would suggest to do chromatin immunoprecipitation with S2 cells or similar, provided they express dpp and they respond to tet-stat92E-bap55 interactions (they use this cell type to search for tet-associated proteins, Fig. 3A). Alternatively, they could try an EMSA (electrophoretic mobility shift assay) with suitable dpp sequences.

Response: We have conducted ChIP-qPCR experiments to show that Tet-C-GFP, Bap55-HA, and Stat92E-HA are directly associated with a previously identified *dpp2.0* enhancer in the *dpp* genomic region, which is known to be required for controlling *dpp* expression in cap cells (Fig. 6). Interestingly, we have also observed a decrease in the enrichment of Bap55-HA and Stat92E-HA pull-down when Tet was knocked down. Our new experimental results show that Tet, Bap55 and Stat92E are indeed co-localized to the *dpp* enhance to control its expression in the niche.

Minor points:

1- The Title of the MS reads "A novel dioxygenase-independent tet function...". I am not sure if this is really a novel function. I have found several published articles in which some of the vertebrate Tet

genes are required to complex transcription factors to their target genes.

Response: In our study, we discovered that Tet acts as a scaffolding protein, connecting the important PBAP complex with Jak/Stat signaling to control *dpp* expression in the *Drosophila* ovarian stem cell niche. This finding is novel and distinct from what has been previously reported in the literature.

2- *Please explain the meaning of IGS. Do you refer to escort cells?*

Response: Inner germarial sheath (IGS) cells are also called escort cells. We updated this in the revised manuscript.

3- *"In the germarium, two or three GSCs directly contact cap cells and anterior IGS cells, which form the niche for controlling their self-renewal, whereas CBs, mitotic cysts and early 16-cell cysts are wrapped up by the long cellular processes of posterior IGS cells, which constitute the niche for promoting these GSC progeny differentiation". Consider improving the grammar. Also, which renewal are you refereeing to? I assume it is GSC renewal, but it is not clear from the sentence.*

Response: The sentence is revised.

4- *"Furthermore, Jak-Stat signaling controls Dpp expression in cap cells to maintain GSC self renewal". Please cite, here and in the rest of the text, Lopez-Onieva et al. Development 2008 (Jak/Stat signalling in niche support cells regulates dpp transcription to control germline stem cell maintenance in the Drosophila ovary).*

Response: The inadvertently missed reference is cited as it should.

5- *"In this study, two independent transgenic shRNA lines were used to knock down Tet in adult cap cells (bab1ts>Tet-KD1 and bab1ts>Tet-KD2)". Careful with this statement (and also in other sections of the text). In addition to cap cells, bab1-Gal4 is also expressed in terminal filament cells and in escort cells.*

Response: Corrected

6- *I would not claim that LamC is "specifically expressed in TF and cap cells". It is enriched in these cell types, while also presenting lower levels in other cells of the germarium.*

Response: Corrected.

7- *Arrowhead in Fig. 1K seems to be pointing to the ring-canal associated fusome material of a 4-cell cyst recently divided, not to a CB spectroosome.*

Response: The image has been replaced.

8- *When referring to the BAC transgenic strain for Bap111-GFP, please specify. Consider stating that Bap111-GFP is an in-frame insertion in the genomic locus of a BAC transgene. Therefore, the flies have the two endogenous bap111 copies plus the BAC's babp111-GFP.*

Response: It has been modified accordingly.

9- *"To determine which complex is involved in controlling dpp expression, we sought to knock down the expression of brm, Bap170, Bap180, and osa in adult niche cells, brm-KD, Bap170-KD,*

Bap180KD, and osa-KD. brm-KD, Bap170-KD, Bap180KD, and osa-KD significantly decrease GSCs compared to the control". Please amend.

Response: Corrected.

10- Fig. 5B: Why does the input Flag-tet-CF-1 have so little amount compared to the other lines? Please explain in the fig. legend, particularly so when the IPed Flag-tet-CF-1 line has plentiful. Why does the IPed Flag-tet-CF-4 line show two bands?

Response: This may be due to the instability of the Flag-Tet-CF-1 protein. However, after enrichment with Flag antibody, we were able to detect a higher amount of Flag-Tet-CF-1, indicating the successfulness of the Co-IP experiment. A previous study reported that Tet proteins can undergo cleavage (Wang, Y. and Y. Zhang, 2014. Cell Reports 6(2): 278-284. PMID: 24412366). The presence of two bands in Flag-tet-CF-4 may be attributed to the generation of a smaller band through protein cleavage, which is enriched by the Flag antibody. Therefore, it is not evident in the Input, but it can be enriched after IP.

11- Fig. 5D, F: why are there so many bands of Myc-Tet-C (D, E) and HA-Stat92E (F)? What do the asterisks mean in D-F?

Response: Asterisks (*s) indicate these bands formed due to protein instability and degradation. We have added a description regarding the asterisks in the figure legend.

12- Bap55-Stat92E fusion: How was this fusion made? Details in the M+M are scarce and confusing, at least to me, as I do not know if it is just an in-frame fusion of both full length ORFs or something different. This is one of the most important experiments of the MS and should be reported with greater detail.

Response: We have added a detailed protocol in the Methods section regarding the generation of the Bap55-Stat92E fusion.

13- Fig. 5G: I assume the difference in the number of GSCs/germarium of wt and tet-KD-1 is statistically significant. The authors may want to show that in the graph, as they have done with 5K and 5L.

Response: It is statistically significant, which has been updated in the figure.

14- Human tet3: It would be interesting to show if a dead version of the Tet3 protein is also able to rescue.

Response: Yes, it would be nice to have the experiment done. Our experiments on the fly Tet has implied that this would be the case. The allowed revision time is not sufficient for us to construct the new transgenic fly strain and conduct genetic crosses.

Referee #3:

This manuscript demonstrates a novel function of TET proteins, independent of catalytic activity, to act as a scaffold protein the associate the PBAP chromatin remodelling complex with STAT92E in

order to activate expression of Dpp in the Drosophila ovarian stem cell niche. This is a highly significant study from 2 aspects: 1) association of a novel TET activity (and also showing conservation of function between Drosophila and human) but also 2) it has been long known that STAT and Dpp signalling somehow interact in the stem cell niche but the mechanism has been unclear. This study provides mechanistic evidence for this interaction. The manuscript is well written and data are clearly presented.

Concerns that need to be addressed:

Response: We appreciate the valuable time and the positive comments.

1) The authors state "The Tet-KD1 and Tet-KD2 germaria significantly decrease GSCs compared to the control although the control and Tet knockdown germaria have comparable CBs (Fig. 1C and 1D)." Why does a loss of GSCs also not flow on the result in a loss of CBs?

Response: This reviewer raised an interesting question. This phenomenon has been observed many times by our lab. We speculate that as the GSC number decreases, the CB might also slow down its differentiation process, including cyst formation.

2) The Materials and Methods state that all of the statistical tests are T-tests. Given that the graphs all show more than two groups should the statistical tests not all have been ANOVAs?

Response: Although there are multiple groups in some of the graphs, we have only done the pairwise comparison, so we only use Student's t-test for our statistical analysis. For example, a RNAi group is compared with the *luc-KD* control group to determine the significance of the change induced by gene knockdown. In the revised manuscript, we have indicated with arrows in the graphs the two groups used for t-tests.

3) The authors state "Like Tet overexpression, niche-specific hTET3 overexpression (hTET3-OE) alone can slightly increase dpp mRNA expression in cap cells and pMad expression in GSCs but does not have obvious effect on GSCs (Fig. 6A-F)." However, Fig. 6F does not show an increase in the hTet3-OE column compared to WT control, and 6D does not show an increase in pMad expression compared to WT.

Response: Corrected.

4) In the discussion could the authors please provide further statements regarding the interaction between the PBAP complex and STAT92E. Does STAT92E need to be prior phosphorylated via JAK activity before interacting with STAT92E.

Response: A statement is provided as suggested.

Additional minor concerns are:

1) In Figure 1B could a germ cell marker also be shown to conclusively show that TET-EGFP is not expressed in germ cells

Response: We use co-staining of Tet-EGFP and Vasa to show that Tet-EGFP is absent in Vasa-positive germ cells.

Referee #1, additional comment:

One point for discussion is the reviewer comment about T-tests vs ANOVA. Although the figures do

show quantification of more than two groups, the relevant statistical tests are the specific pairwise comparisons between wildtype vs each of the experimental genotypes. ANOVA would be more appropriate if the experimenters had no a priori knowledge of which pairwise comparison would be most relevant or interesting (e.g. a difference between two of the experimental genotypes would be just as important as a difference between a control and an experimental). Since this is not the case, I believe T-tests are most appropriate. However, since they are performing multiple T-tests in the same experiment, one could argue that a multiple comparison correction such as bonferroni should be applied, but I don't think that is commonly done in this case.

Response: Thank you very much for this comment! We have addressed this issue earlier accordingly.

Dear Dr Ting Xie,

Thank you for submitting your revised manuscript (EMBOJ-2023-115586R) to The EMBO Journal. Your amended study was sent back to the three referees for their scientific re-evaluation, and we have received detailed comments from all of them, which I enclose below.

As you will see, the experts state that the work has been substantially improved by the revisions and they are now in favour of publication, pending minor revision.

Thus, we are pleased to inform you that your manuscript has been accepted in principle for publication in The EMBO Journal.

Please consider the remaining minor comment of referee #2 regarding sample annotation carefully and amend the manuscript text accordingly.

Also, we now need you to take care of a number of issues related to formatting and data presentation as detailed below, which should be addressed at re-submission.

Please contact me at any time if you have additional questions related to below points.

Thank you for giving us the chance to consider your manuscript for The EMBO Journal. I look forward to your final revision.

Again, please contact me at any time if you need any help or have further questions.

Best regards,

Daniel Klimmeck

>> Author Contributions: Please remove the author contributions information from the manuscript text. Note that CRediT has replaced the traditional author contributions section as of now because it offers a systematic machine-readable author contributions format that allows for more effective research assessment. and use the free text boxes beneath each contributing author's name to add specific details on the author's contribution.

More information is available in our guide to authors.
<https://www.embopress.org/page/journal/14602075/authorguide>

>>Manuscript composition: the order of the manuscript sections should be: ...References / Figure legends / Expanded View Figure legends.

>> Appendix figure legends should be removed from the manuscript file.

>> Please indicate redisplay of the 'luc-KD' data in Figure 1E in the figure legend for Figure 4C.

>> Adjust the title of the 'Declaration of Interests' section to 'Disclosure and Competing Interests Statement'.

>> Remove the 'Peer Review Information' paragraph from the manuscript.

>> Revisit publication status of the bioRxiv reference Singh et al (2023) and update in case of formal journal publication.

>> Author checklist: change the information provided on "Data Availability >>Primary Datasets provided" to 'No" (this applies

only to global datasets deposited in public databases).

>> Consider additional changes and comments from our production team as indicated below:

- Figure Legends (main + EV):

Please note that in figure 6f; there is a mismatch between the annotated p values in the figure legend and the annotated p values in the figure file that should be corrected.

Referee #1:

With this revised manuscript, the authors have fully addressed my previous comments and I now support publication.

Referee #2:

The AUs have addressed most of my concerns and my major criticism (the binding of the Stat-Tet-Bap55 complex to the dpp gene) satisfactorily.

In the new quantification by qPCR of the efficiency of the RNAi approach, the AUs ought to explain which tissue they isolated the mRNA from. I assume it was from ovary tips, as the reduction in mRNA levels was quite significant for all the genes, and using the bab1-Gal4 line to drive the shRNA would only affect early oogenesis.

Referee #3:

I am now satisfied that the authors have made substantial improvements to the manuscript and addressed all of my concerns.

The authors addressed the remaining editorial issues.

Dear Dr Ting Xie,

Thank you for submitting the revised version of your manuscript. I have now evaluated your amended manuscript and concluded that the remaining minor concerns have been sufficiently addressed.

I am pleased to inform you that your manuscript has been accepted for publication in the EMBO Journal.

On a different note, I would like to alert you that EMBO Press offers a format for a video-synopsis of work published with us, which essentially is a short, author-generated film explaining the core findings in hand drawings, and, as we believe, can be very useful to increase visibility of the work. This has proven to offer a nice opportunity for exposure i.p. for the first author(s) of the study. Please see the following link for representative examples and their integration into the article web page:

<https://www.embopress.org/doi/full/10.15252/embo.2019103932>

If you have any questions, please do not hesitate to contact the Editorial Office.

Best regards,

Daniel Klimmeck

Daniel Klimmeck, PhD
Senior Editor
The EMBO Journal
EMBO
Postfach 1022-40
Meyerohofstrasse 1
D-69117 Heidelberg
contact@embojournal.org
Submit at: <http://emboj.msubmit.net>
